# pHmScarlet is a pH-sensitive red fluorescent protein to monitor exocytosis docking and fusion steps

Anyuan Liu [1,2,9], Xiaoshuai Huang [3,9], Wenting He[2], Fudong Xue[2], Yanrui Yang[4], Jiajia Liu [4,5], Liangyi Chen [6], Lin Yuan [2✉] & Pingyong Xu [1,2,5,7,8✉]

pH-sensitive fluorescent proteins (FPs) are highly advantageous for the non-invasive monitoring of exocytosis events. Superecliptic pHluorin (SEP), a green pH-sensitive FP, has been widely used for imaging single-vesicle exocytosis. However, the docking step cannot be visualized using this FP, since the fluorescence signal inside vesicles is too low to be observed during docking process. Among the available red pH-sensitive FPs, none is comparable to SEP for practical applications due to unoptimized pH-sensitivity and fluorescence brightness or severe photochromic behavior. In this study, we engineer a bright and photostable red pH-sensitive FP, named pHmScarlet, which compared to other red FPs has higher pH sensitivity and enables the simultaneous detection of vesicle docking and fusion. pHmScarlet can also be combined with SEP for dual-color imaging of two individual secretory events. Furthermore, although the emission wavelength of pHmScarlet is red-shifted compared to that of SEP, its spatial resolution is high enough to show the ring structure of vesicle fusion pores using Hessian structured illumination microscopy (Hessian-SIM).

[1] School of Life Sciences, University of Science and Technology of China, Hefei, Anhui, China. [2] Key Laboratory of RNA Biology, Institute of Biophysics, Chinese Academy of Sciences, Beijing, China. [3] Biomedical Engineering Department, Peking University, Beijing, China. [4] State Key Laboratory of Molecular Developmental Biology, Institute of Genetics and Developmental Biology, Chinese Academy of Science, Beijing, China. [5] College of Life Sciences, University of Chinese Academy of Sciences, Beijing, China. [6] State Key Laboratory of Membrane Biology, Beijing Key Laboratory of Cardiometabolic Molecular Medicine, Institute of Molecular Medicine, Peking University, Beijing, China. [7] National Laboratory of Biomacromolecules, Institute of Biophysics, Chinese Academy of Sciences, Beijing, China. [8] Department of Clinical Laboratory, Children's Hospital of Chongqing Medical University, Chongqing, China. [9]These authors contributed equally: Anyuan Liu, Xiaoshuai Huang. ✉email: yuanlin@ibp.ac.cn; pyxu@ibp.ac.cn

Vesicle exocytosis is a rapid dynamic process that controls the release of vesicle encapsulated active peptides or neurotransmitters to the exterior of the cell, thereby regulating important physiological functions[1,2]. This process is mainly monitored using electrophysiological and optical imaging methods. In particular, the patch-clamp electrophysiological technique can effectively detect the total exocytosis and/or endocytosis signals of all vesicles in a single cell at very high temporal resolution. Unfortunately, this technique is invasive and cannot be easily used to assess single-vesicle exocytosis[3]. Optical imaging techniques, in contrast, allow for the assessment of individual vesicle exocytosis in single or multiple cells, which is essential for the elucidation of fundamental exocytosis mechanisms. Total internal reflection fluorescence microscopy (TIRFM) is an optical technique that is commonly used to detect single-vesicle exocytosis (~100 nm below the plasma membrane) by analyzing the changes in the fluorescence intensity of a pH-sensitive fluorescent protein (FP; pKa >6)[4]. FPs are generally used to label the vesicular membrane protein (or other content) and are localized in the acidic vesicles (pH 5.5). Upon release to the extracellular environment (pH 7.4), the fluorescence intensity of FPs increases. This effect is directly related to the pH sensitivity of each FP, a property that is controlled by two main factors: pKa (pH value at which the fluorescence intensity is 50% of maximum) and the apparent Hill coefficient ($n_H$) (the slope of the fluorescence versus pH curve)[5]. An ideal FP is characterized by an optimal pKa value of 7.5 and a high value of $n_H$. Among the available FPs, SEP has a nearly optimal pKa (7.2) and the highest $n_H$ (1.90)[5]. As such, it is highly pH sensitive, and it shows the greatest change in fluorescence intensity upon transfer to the extracellular environment. Therefore, SEP is considered to be the most suitable FP for the detection of exocytosis events.

The combination of SEP and a red pH-sensitive FP enables dual-color imaging of two different membrane cargos on a single vesicle or on two types of vesicles in the same cell. Among the currently available red and orange FPs (e.g., mOrange2[6], mNectarine[7], pHTomato[8], pHred[9], pHoran[5], pHuji[5], etc.), pHuji has the highest $n_H$ value and the most optimal pKa in the physiological range[5]. However, the pH sensitivity of this FP is significantly lower than that of SEP. In addition, pHuji is characterized by low intrinsic fluorescence intensity at pH 7.5 and minimal burst signal of fusion, which limits its application in the detection of exocytosis[10]. pHuji also exhibits pronounced photo-switching behavior wherein its fluorescence decreases to 35% of the initial value in less than 2 s of continuous illumination[5]. This renders the protein unsuitable for quantitative imaging of exocytosis.

Several complex molecular mechanisms are implicated in vesicle exocytosis. In general, the process occurs in three main steps: vesicle docking, priming, and fusion[11]. Typically, the docked vesicles that appear near the plasma membrane are visualized using an electron microscope[12]. However, this instrument cannot be used to simultaneously detect the dynamic stages of docking and fusion in living cells. Although SEP is ideal for the detection of vesicle fusion, its low fluorescence intensity does not change during docking since the pH remains the same[13]. Contrarily, EGFP-labeled vesicles exhibit strong fluorescence in the docking stage; however, their burst fusion signal is not very high[14]. Therefore, EGFP is not suitable for the detection of single-vesicle fusion events, especially when the secreted vesicle is fused to the plasma membrane and produces very high background fluorescence. Likewise, pHuji cannot be used to reliably monitor the docking and fusion kinetics of vesicle exocytosis due to its photo-switching behavior[5].

Considering that fusion pore formation is highly dynamic, and that it occurs at the nanometer scale within just a few milliseconds, the process can only be observed using an FP with a high fusion signal combined with an ultrafast super-resolution (SR) imaging technique. Previously, stimulated emission depletion (STED) microscopy and membrane-labeled dyes have been successfully applied in the visualization of fusion pore formation[15]. Huang et al. were able to observe enlarged fusion pores using SEP and quick live-cell Hessian-SIM microscopy[16]. As for the red pH-sensitive FPs, it is more challenging to observe fusion pores due to the decrease in resolution upon the red shifting of the FP emission wavelength (Abbe and Rayleigh criteria)[17]. To the best of our knowledge, the use of red pH-sensitive FPs for the detection of fusion pores has not been reported.

In this study, we propose a photostable red pH-sensitive FP (pHmScarlet) that is developed from mScarlet-I for the simultaneous detection of vesicle docking and fusion. Compared to other red FPs, pHmScarlet has the highest pH sensitivity, and its fluorescence intensity in living cells is seven times greater than that of pHuji. Furthermore, when combined with the Hessian-SIM analytical technique, pHmScarlet can be used to observe fusion pore formation during exocytosis at high spatial resolution.

## Results

**Development of pHmScarlet**. Considering its fast maturation and the fact that its emission peak is well separated from that of SEP, mScarlet-I[18], the brightest available red FP, was chosen as a template for the development of a bright red pH-sensitive FP with optimum pKa and $n_H$ values. With a pKa of 5.4, mScarlet-I is not appreciably sensitive to changes in pH during vesicle secretion, and thus, it cannot be used to detect vesicle exocytosis. In a previous study, it had been shown that residue 163 in pHuji, which corresponds to residue M164 in mScarlet-I, increases the pKa of mApple[5]. Therefore, this residue was selected for saturation mutation, and the fluorescence intensity of the mutated FP was measured at pH 5.5 and 7.5. The pKa of the identified M164K mutant (named mScarlet-IK) was found to be 6.8, and the difference between the mutant's fluorescence intensities determined at 5.5 and 7.5 pH conditions was shown to be 16 folds. Based on the structure of mScarlet[18], we speculated that the two amino acids (K163 and A165) adjacent to M164K may affect the interaction of this residue with the chromophore through S147, thereby altering its pH sensitivity. To test this hypothesis, saturation mutagenesis was carried out at residues 163 and 165. As expected, the mScarlet-IKLV mutant (K163L and A165V) termed pHmScarlet0 (pH-sensitive mScarlet0) presents increased pH sensitivity compared to mScarlet-IK (Supplementary Fig. 1). The maxima of absorbance and emission of pHmScarlet0 is 564 and 589 nm, respectively (Supplementary Fig. 2). The pKa of pHmScarlet0 is 7.2, and it shows a 20-fold change in fluorescence intensity upon increasing the pH from 5.5 to 7.5 (Table 1).

In order to further improve pH sensitivity, all residues interacting with the chromophore in pHmScarlet0 (residues 71, 94, 96, 110, and 147) were subjected to saturation mutagenesis. However, based on the results of fluorescence experiments, most of these mutants fluoresce less upon mutation. Considering that the residues adjacent to the groups that can directly interact with the chromophore (residues 64, 112, 178, and 198) may also influence pH sensitivity, the effect of mutation on the fluorescence behavior of these residues was also tested.

By mutating and testing different combinations of residues, we were finally able to develop a bright ecliptic variant that has the highest biological pH sensitivity, with ~26-fold change in fluorescence intensity upon increasing pH from 5.5 to 7.5 (Table 1, Fig. 1a). As for the pH-dependent absorbance spectra, a clear increasement can be observed in the blue absorption below pH 7, representing the protonated chromophore (Fig. 1b). To the best of our knowledge, this is the largest fold change ever detected

**Table 1 In vitro characterization of pH-sensitive FPs.**

| | pKa | $n_H$ | Fluorescence fold change (pH 5.5-7.5) | Excitation peak at pH 7.5 (nm) | Emission peak at pH 7.5 (nm) | Extinction coefficient ($10^3 M^{-1} cm^{-1}$) | QY (-) | Brightness ($10^3 M^{-1} cm^{-1}$) | $t_{1/2}$ (s) |
|---|---|---|---|---|---|---|---|---|---|
| SEP | 7.2 | 1.9 | 50 | 495 | 512 | 45 | 0.52 | 23.15 | - |
| pHuji | 7.7 | 1.0 | 22 | 572 | 598 | 31 | 0.22 | 6.82 | 1 |
| pHmScarlet0 | 7.2 | 0.9 | 20 | 564 | 589 | 86 | 0.48 | 41.53 | 53 |
| pHmScarlet | 7.4 | 1.1 | 26 | 562 | 585 | 85 | 0.47 | 39.73 | 28 |

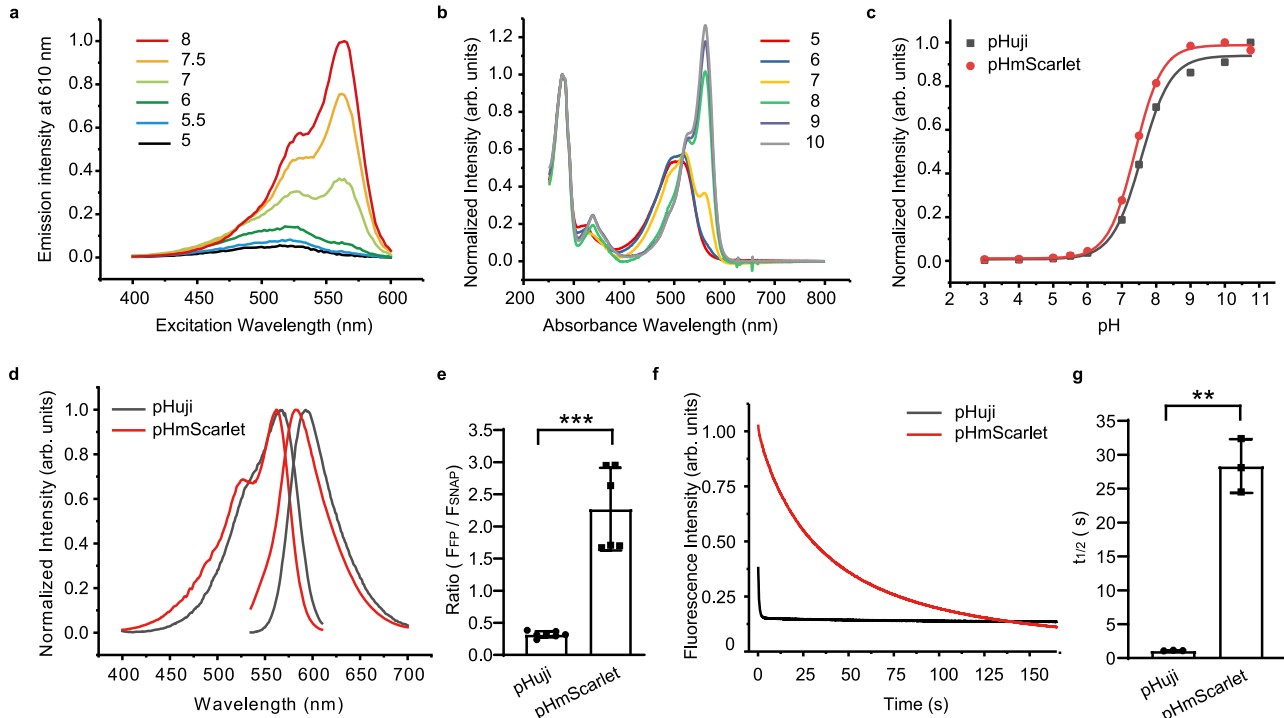

**Fig. 1 Characteristics of pHmScarlet. a** Excitation spectra and **b** Absorbance spectra of pHmScarlet at different pH values. **c** pH titration curves of pHmScarlet (red) and pHuji (black). **d** Excitation and emission spectra of pHmScarlet (red) and pHuji (black) at pH 7.4. **e** SNAP-normalized fluorescence of FPs in U-2 OS cells expressing H2B-pHmScarlet or H2B-pHuji (pHuji: 0.32 ± 0.05, pHmScarlet: 2.27 ± 0.64, $p = 0.0007$, two-sided Student's $t$ tests. Data are presented as mean ± SD). **f** Photobleaching curves of pHmScarlet (red) and pHuji (black). **g** $t_{1/2}$ of pHmScarlet and pHuji (pHuji: 1.1 ± 0.1, pHmScarlet: 28.3 ± 2.5, $p = 0.007$, two-sided Student's $t$ tests. Data are presented as mean ± SD). Results shown are from three independent experiments. **$^{**}p$ <0.01, $^{***}p$ <0.001. Source data is provided as a source data file.

among red FPs in this pH range. The variant is labeled pHmScarlet, and it is characterized by T148C, K199Q, and D201T mutations (Supplementary Fig. 1). Notably, the pKa value of pHmScarlet is nearly optimal (7.4), and the FP has the same apparent $n_H$ as pHuji (Table 1, Fig. 1c). pHmScarlet fluoresces at approximately 585 nm (Table 1, Fig. 1d), and thus, its emission peak is well separated from that of SEP (512 nm; Table 1). Also, purified pHmScarlet (extinction coefficient [EC] of 85,000 $M^{-1}$ $cm^{-1}$ and quantum yield of 0.47) is almost six times brighter than pHuji (Table 1). In living U-2 OS cells, the variation in brightness between the two FPs increases to nearly sevenfold, as evidenced by the results of SNAP-tag analysis illustrated in Fig. 1e. Finally, pHmScarlet is more photostable than pHuji, and it does not exhibit apparent photo-switching behavior, as shown in the photobleaching curves presented in Fig. 1f. Under the same imaging conditions, the time point where the fluorescence intensity was decreased to 50% of the initial fluorescence intensity ($t_{1/2}$)[18] of pHmScarlet is 28 s, compared to 1 s for pHuji (Fig. 1g). We further examined the photochromic behavior of pHmScarlet

by alternately exciting the FPs with 561 nm and 488/405 nm lights. No fast photo-switching was measured for pHmScarlet (Fig. 2a, d, e, h) and mScarlet (Fig. 2b, d, f, h), while pHuji showed severe photochromic behavior of 34.4% at 488 nm (Fig. 2c, d) and 36% at 405 nm (Fig. 2g, h).

To verify whether pHmScarlet0 and pHmScarlet are true monomeric FPs, the organized smooth endoplasmic reticulum (OSER) approach was performed[18,19]. For better determination of OSER structures, EGFP was included as a positive control for dimerization, and mScarlet was included as a negative control as it is stated to be a true monomer[18]. Results showed that similar as mScarlet (Fig. 3a, f), pHmScarlet0 (Fig. 3b, f), and pHmScarlet (Fig. 3c, f) proved to be monomeric in U-2 OS cells, with over 89% of cells showing normal phenotypes. The averaged intensity ratio of the OSER structure over the mean intensity of the nuclear envelope (NE) of pHmScarlet0 and pHmScarlet was comparable to that of the monomeric mScarlet and much lower than that of EGFP (Fig. 3e, f). Consistent with their high brightness and monomeric behavior, pHmScarlet0 and pHmScarlet perform well in labeling

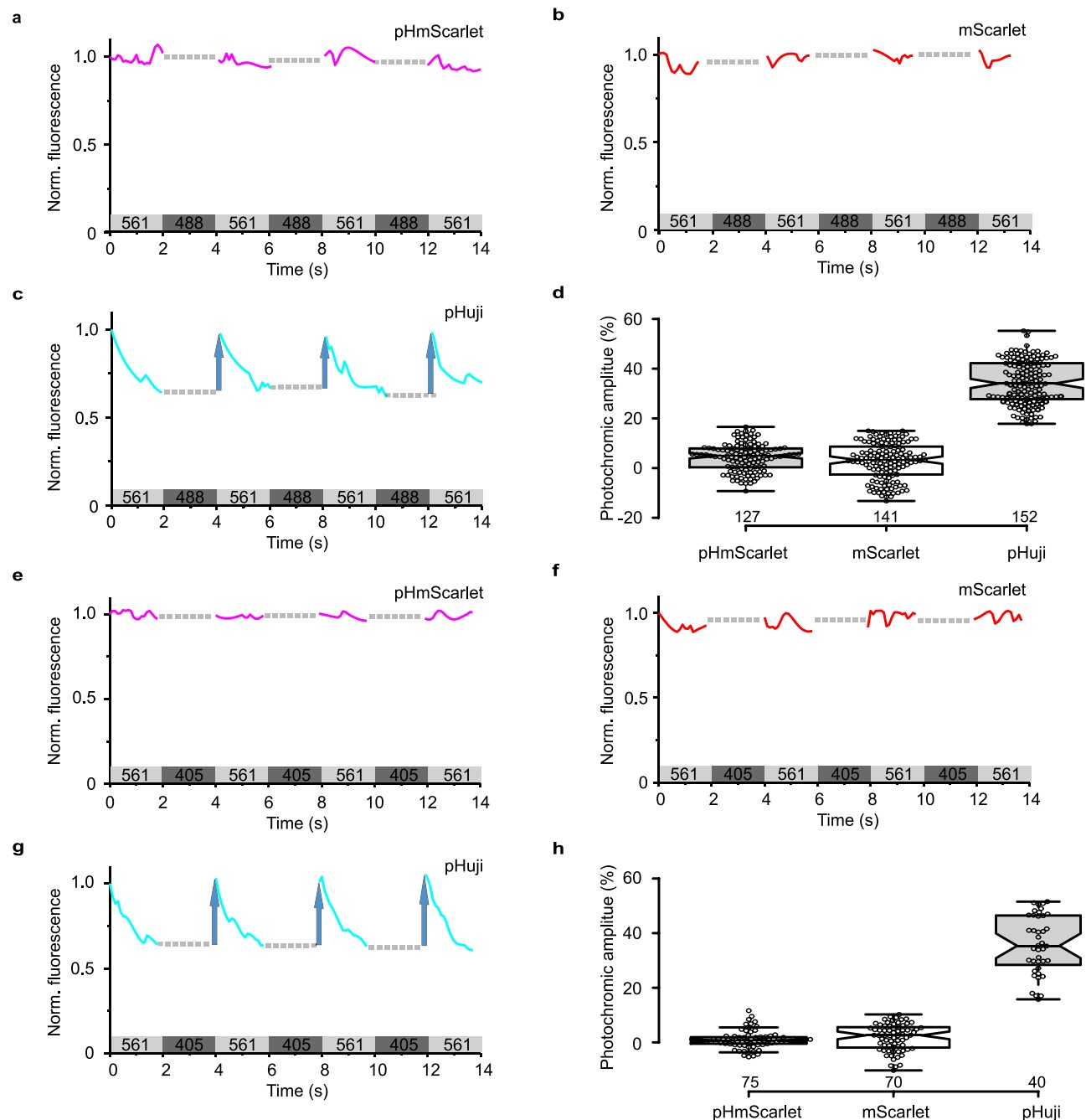

**Fig. 2 Photochromic behavior of RFPs in U-2 OS cells.** The cells were widefield illuminated with alternating light of 561 nm and 488 nm (**a–c**) or 405 nm (**e–g**) lights for multiple illumination cycles. The photochromic amplitudes (blue arrows) of RFPs determined for each cycle are presented in (**d**) and (**h**). Boxplots were generated using BoxPlotR (http://boxplot.tyerslab.com/), which includes the median values, the individual data points, and the 95% confidence intervals (notches). **d** $n = 14$ cells for pHmScarlet, $n = 14$ cells for mScarlet, and $n = 19$ cells for pHuji examined. Independent experiments were repeated twice for each FPs. The median, upper whisker, and lower whisker are 5%, 17%, and −9% for pHmScarlet, 3%, 15%, and −13% for mScarlet, and 34%, 55%, and 18% for pHuji, respectively. **h** $n = 5$ cells for pHmScarlet, $n = 7$ cells for mScarlet, and $n = 4$ cells for pHuji. Independent experiments were repeated three times for each FPs. The median, upper whisker, and lower whisker are 1%, 5%, and −4% for pHmScarlet, 3%, 10%, and −10% for mScarlet, and 35%, 52%, and 16% for pHuji, respectively. **a**, **e**: pHmScarlet; **b**, **f**: mScarlet; **c**, **g**: pHuji. Source data is provided as a source data file.

α-tubulin. Similar as that of mScarlet (Fig. 3g), expression of pHmScarlet0 or pHmScarlet-tagged α-tubulin resulted in bright microtubules with minor background fluorescence (Fig. 3h, i), compared to the high background fluorescence of EGFP (Fig. 3j). Moreover, to further prove there is no disulfide-bridge formation between molecules, we checked the protein mobility on nonreducing SDS-PAGE under oxidative and reducing environment. As shown in Supplementary Fig. 3, purified mScarlet, pHmScarlet0, and pHmScarlet migrated similarly under oxidative and reducing environment (Supplementary Fig. 3a). However, the purified Ero1α, included as a positive control, migrated faster under oxidative treatment (Supplementary Fig. 3b).

**Detection of single-vesicle exocytosis using pHmScarlet.** The effectiveness of pHmScarlet in detecting single-vesicle exocytosis was assessed using living INS-1 cells that secrete insulin-

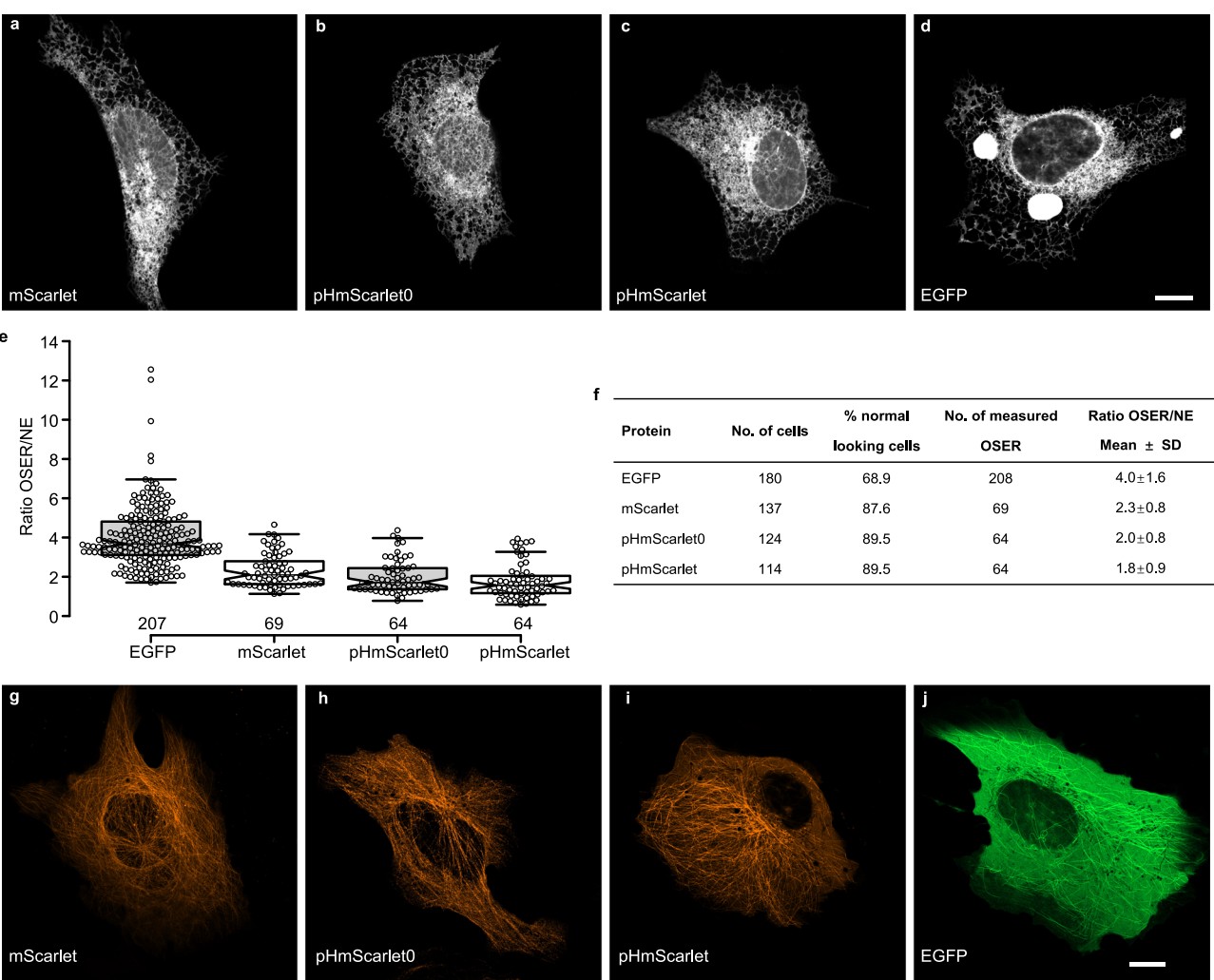

**Fig. 3 Assessment of monomeric properties of FPs in U-2 OS cells. a–d** OSER assay of U-2 OS cells transfected with plasmids encoding CytERM-FPs: **a** mScarlet, **b** pHmScarlet0, **c** pHmScarlet, and **d** EGFP. Scale bar is 10 μm. **e** Ratio of the averaged intensity of the OSER over the mean intensity of the NE. Each dot represents an OSER structure. The median, upper whisker, and lower whisker are 3.68, 6.96, and 1.70 for EGFP, 2.10, 4.17, and 1.13 for mScarlet, 1.71, 3.97, and 0.78 for pHmScarlet0, and 1.56, 3.27, and 0.58 for pHmScarlet, respectively. **f** Table displaying the results of the OSER assay. 'Normal looking cells' are cells without OSER structures and without incorrect localization. **g–j** Microtubule structures in U-2 OS cells transfected with FP-7aa-α-tubulin: **g** mScarlet-α-tubulin, **h** pHmScarlet0-α-tubulin, **i** pHmScarlet-α-tubulin, and **j** EGFP-α-tubulin. Experiments were repeated two times for each FP. Scale bar is 10 μm. Source data is provided as a source data file.

containing vesicles upon high glucose and high $[K^+]$ stimulation. The vesicle-specific membrane protein VAMP2 (vesicle-associated membrane protein 2) was labeled with pHmScarlet, SEP, or pHuji, then the fluorescence signals originating from single vesicles were measured under different conditions.

The results illustrated in Fig. 4a and Supplementary Movie 1 show that the exocytosis of stimulated single vesicles labeled with pHmScarlet is detected in the form of a sudden burst in fluorescence. This burst is attributed to the transfer of pHmScarlet from the acidic intra-vesicular environment to the weakly basic extracellular medium. The vesicles labeled with SEP and pHuji also exhibit fluorescence bursts upon stimulation (Fig. 4c, e). For all probes, the peak fluorescence intensity varies between different vesicles. Compared to SEP and pHuji, pHmScarlet exhibits a wider distribution of fluorescence bursts ($\Delta F_{max}/F_0$, $\Delta F_{max} = F_{max}-F_0$; $F_{max}$ is the maximum fluorescence intensity of the peak of exocytosis, $F_0$ is the background fluorescence intensity of the vesicle immediately before secretion), and the number of pHmScarlet-labeled vesicles having $\Delta F_{max}/F_0 > 3$ is greater (Fig. 4b,

d, f). As pHmScarlet produce lower background on the cell surface than SEP, and is much brighter than pHuji, it has higher S/N (signal to noise ratio) than SEP and pHuji for the detection of single-vesicle exocytosis in cells. VAMP2-pHmScarlet and VAMP2-SEP present similar profiles of averaged exocytotic kinetics (Supplementary Fig. 4), which indicates that pHmScarlet can be used as a functional probe for vesicle labeling and single-vesicle exocytosis detection. A comparison of the total numbers of individual exocytosis events detected in a single cell using the three probes shows that more exocytotic vesicles can be detected using pHmScarlet and SEP than using pHuji. Therefore, the detection efficiency of red pHmScarlet FP is similar to that of SEP, but greater than that of pHuji (Fig. 4g).

To test whether pHmScarlet can efficiently detect the exocytosis of synaptic vesicles in neuronal cells, VAMP2-pHmScarlet was transiently expressed in the HT-22 mouse hippocampal neuronal cell line, then vesicle exocytosis was evoked by high $[K^+]$ stimulation. The fluorescence profile presented in (Fig. 4h and Supplementary Movie 2) shows a rapid

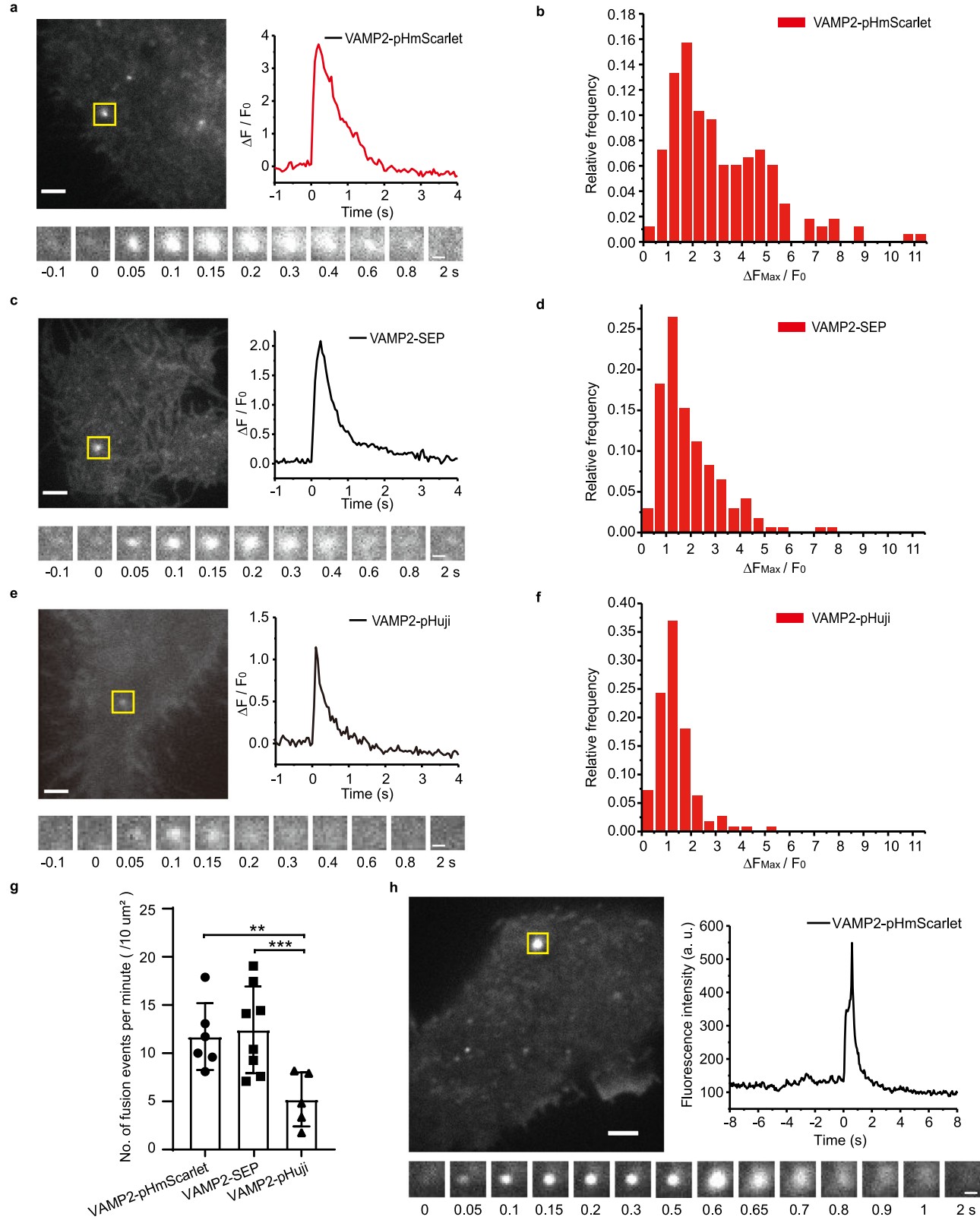

initial rise in fluorescence intensity due to the opening of stimulated fusion pores, followed by a fluorescence decay due to the diffusion of VAMP2-pHmScarlet in the plasma membrane. When co-expressing with SEP and pHmScarlet or SEP and pHuji, the ability to detect individual exocytotic events labeled by pHmScarlet was comparable to that of SEP (Supplementary Fig. 5). In primary mouse hippocampal pyramidal neurons, single

exocytotic fusion events can be detected by VAMP2-pHmScarlet in the soma region of these cells (Supplementary Fig. 6).

**Detection of vesicle docking using pHmScarlet**. As the vesicles approach the plasma membrane at the upstream of the final fusion event (docking stage), their mobility becomes highly

**Fig. 4 Detection of single exocytosis events in INS-1 cells and hippocampal neuronal cells.** TIRF images of single-vesicle exocytosis in INS-1 cells expressing **a** VAMP2-pHmScarlet, **c** VAMP2-SEP, **e** VAMP2-pHuji, and **h** in HT-22 mouse hippocampal neuronal cells expressing VAMP2-pHmScarlet. Fusion events in a single cell are indicated by yellow rectangles (left top), scale bars: 2 μm. Time-lapses of the marked exocytotic events are shown on the bottom (1.3 × 1.3 μm), scale bars: 0.5 μm; normalized intensity traces of the marked events are highlighted as green or red channels (right top). Experiments were repeated five times for pHmScarlet in INS-1 cells (**a**) and three times in HT-22 cells (**h**), three times for SEP (**c**), and three times for pHuji (**e**) in INS-1 cells. $\Delta F_{max}/F_0$ distribution of individual exocytotic events for vesicles labeled with (**b**) VAMP2-pHmScarlet, (**d**) VAMP2-SEP, and (**f**) VAMP2-pHuji. **g** Total number of exocytotic fusion events normalized to the cell area and imaging time, experiments were repeated three times (pHmScarlet: 11.7 ± 3.5, SEP: 12.4 ± 4.5, pHuji: 5.2 ± 2.8, pHmScarlet compared with pHuji, $p = 0.0074$; SEP compared with pHuji, $p = 0.0045$, two-sided Student's t tests. Data are presented as mean ± SD, $^{**}p <0.01$, $^{***}p <0.001$). Source data is provided as a source data file.

restricted. This is evidenced by changes in the fluorescence intensities of the labeled vesicles. Before exocytosis, SEP is almost invisible in the lumen of vesicles due to its very high $n_H$ value and nearly optimal pKa (Fig. 4c). Therefore, this FP is not suitable for the detection of vesicle docking. Conversely, the docking of pHmScarlet-labeled vesicles at the target membrane can be clearly detected in the form of enhanced fluorescence prior to the sudden burst signal of the fusion step (Fig. 5b, c). The dwell times of the docked vesicles vary in the range of 2–30 s, and the distribution is exponential (Fig. 5d). It should be noted that 83.6% of the pHmScarlet-labeled vesicles were found to be docked at the plasma membrane. The remaining vesicles skip the docking stage and enter the fusion step directly (Fig. 5a).

To confirm that the increase in fluorescence prior to fusion is attributed to vesicle docking, INS-1 cells were co-transfected with VAMP2-pHmScarlet and VAMP2-SEP or VAMP2-pHmScarlet and VAMP2-EGFP (the pH sensitivities of SEP and EGFP are higher and lower than that of pHmScarlet, respectively). Then, dual-color imaging was performed in order to detect the changes in the fluorescence intensity of a particular vesicle upon stimulation. The obtained results confirm that many individual vesicles in the red channel exhibit enhanced fluorescence shortly before the burst signal. Such increase in fluorescence intensity is not detected in the green channel of SEP (Fig. 5e, arrow). In contrast, transfection with VAMP2-pHmScarlet and VAMP2-EGFP leads to a notable pre-burst surge in fluorescence that is detected in both, the red and green channels (Fig. 5f, arrow). The increased fluorescence is attributed to the attachment of vesicles to the plasma membrane, and thus, it is indicative of vesicle docking. Importantly, the normalized burst signals of pHmScarlet and SEP (2.54 ± 0.82 and 2.54 ± 0.88, respectively) in cells co-expressing VAMP2-pHmScarlet and VAMP2-SEP are comparable, as are the decay kinetics of the two FPs (Fig. 5e). In contrast, the burst signal of EGFP is significantly weaker than that of pHmScarlet (Fig. 5f) due to the low pH sensitivity and pKa value of the former. Compared to pHmScarlet and SEP, pHuji labeling leads to lower fluorescence intensities, unless the exocytosis signal is normalized to the background ($F_0$), in which case the intensity of pHuji fluorescence is comparable to that of SEP (Fig. 5g). This is probably due to the photobleaching of the diffused pHuji signal after fusion, which results in much lower background fluorescence. However, unlike VAMP2-pHmScarlet (Fig. 5e), VAMP2-pHuji decays more quickly than SEP (Fig. 5g) due to its photo-switching behavior, and so, it is less suitable for the detection of fusion dynamics. Furthermore, a single vesicle co-labeled with VAMP2-pHuji and VAMP2-SEP exhibits two peaks in the red channel, the second of which corresponds to the fusion step as indicated by SEP (Fig. 5g). We quantitatively analyzed the proportion of secreted vesicles that have only one exocytotic event but with a first peak prior (Fig. 5g, arrow) to the fusion peak. We found that 39% of the single exocytotic events were with a small peak, suggesting that nearly 40% of the single exocytotic events are hard to be determined whether it corresponds to a docking event or another fusion event at the same loci when

pHuji were used for exocytosis detection. Combined with that 83% of secreted vesicles are with docking steps when co-labeling vesicles with pHmScarlet and SEP, the data also indicates that nearly 44% of the docking events are missed and cannot be detected with pHuji. These results suggest that pHuji cannot be used to reliably monitor the docking step.

As mentioned earlier, approximately 16% of the secreted vesicles skip the docking stage and proceed directly to the fusion event. There exist two possible explanations for such behavior: either the docking process of these vesicles is too fast to be captured, or their intra-environment is too acidic (more so than the vesicles showing docking steps), and thus, the fluorescence of VAMP2-pHmScarlet cannot be monitored. Considering that the docking-free secretion events recorded using VAMP2-pHmScarlet can also be observed in the green channel of the VAMP2-EGFP-labeled vesicles (the pKa of EGFP is 6.0, Fig. 5h), and that the distribution of the docking dwell times is exponential (Fig. 5d), it is possible that the docking step in docking-free secretions is too fast to be captured. However, the other explanation cannot be entirely dismissed. Further investigation of the molecular mechanism of docking-free exocytosis is needed in order to determine which explanation is true.

**Dual-color imaging of the exocytosis of distinct vesicle membrane cargos using pHmScarlet and SEP.** Having demonstrated that pHmScarlet and SEP are both capable of detecting fusion events, we next investigated their utility in monitoring the exocytosis of two distinct vesicle membrane cargos. For this purpose, INS-1 cells were co-transfected with synaptophysin-pHmScarlet (Syp-pHmScarlet) and VAMP2-SEP, then they were stimulated by increasing the concentrations of glucose and $K^+$. The images presented in Fig. 6a, b show that the stimulated cells exhibit simultaneous fluorescence bursts in the green and red channels of the same vesicle. The similarity between the bursts in the two channels indicates that the response of pHmScarlet to vesicle fusion is akin that of SEP (Fig. 6c). Furthermore, with pHmScarlet the docking steps can be observed (Fig. 6c) and there is 64% of secreted vesicles are with the docking steps. By analyzing the kinetics of fluorescence decay after fusion pore opening, the diffusion of the two cargos in the plasma membrane can be tracked. The results illustrated in Supplementary Fig. 7 show that the averaged fluorescence signals of pHmScarlet and SEP exhibit similar decay kinetics, which means that the two labeled cargos diffuse similarly in the plasma membrane.

**Hessian-SIM super-resolution imaging of enlarged fusion pores using pHmScarlet.** To test whether pHmScarlet can be used for the imaging of fusion pores at super-resolution, VAMP2-pHmScarlet overexpression was induced in INS-1 cells. The stimulation-triggered exocytosis of vesicles in these cells was continuously recorded using Hessian-SIM (frame rate = 97 Hz). Rolling SIM reconstruction was also applied in order to increase the temporal resolving power of the analytical system[16]. The

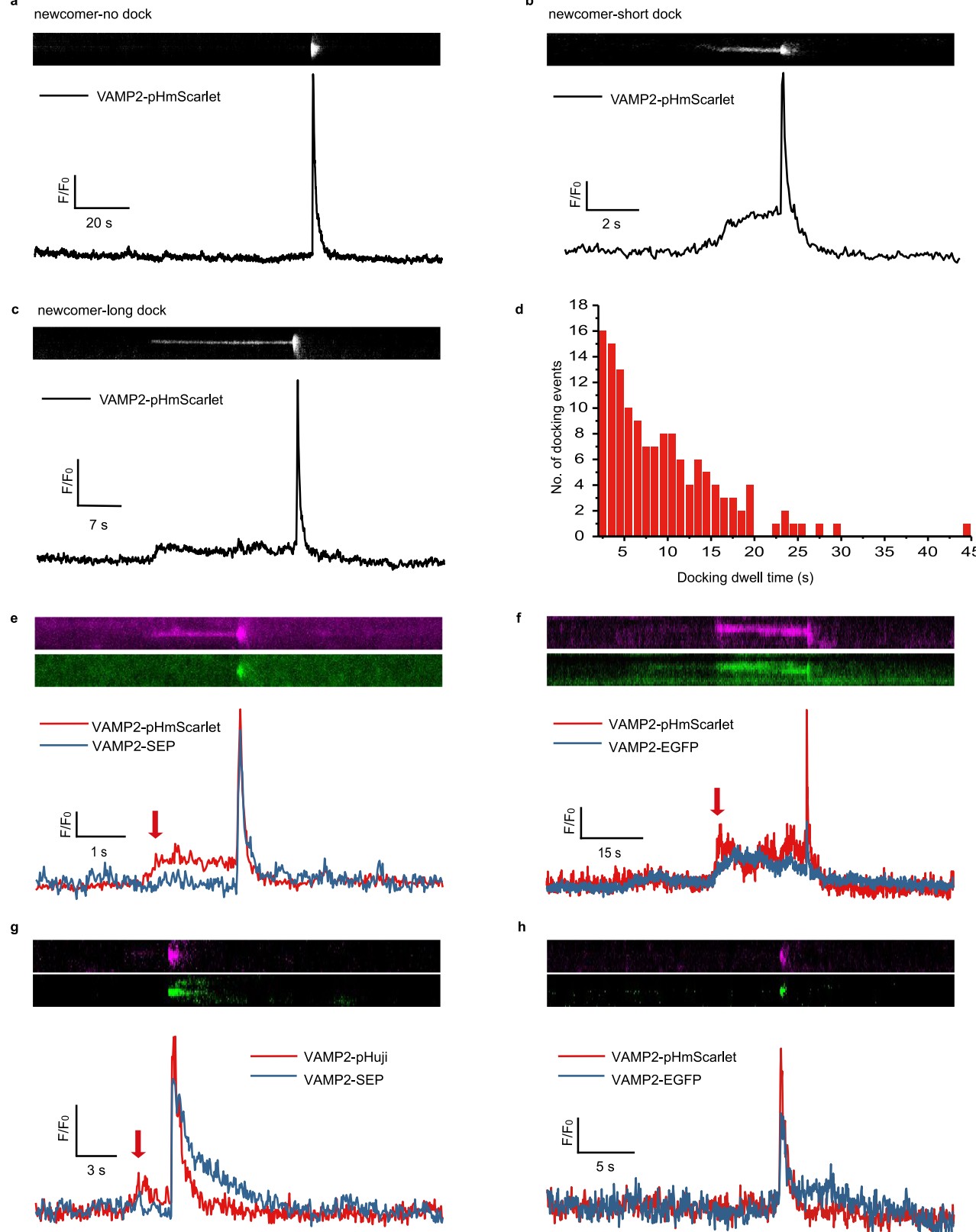

**Fig. 5 Detection of the docking stage in INS-1 cells.** Representative kymograph and the corresponding time-lapse fluorescence-intensity curves showing the distinct kinetic properties of **a** docking-free, **b** short docking, and **c** long docking exocytotic events. **d** Distribution of docking dwell times (total exocytotic evens, $n = 138$; three cells were examined over two independent experiments). Representative kymograph and corresponding time-lapse fluorescence-intensity curves showing the exocytosis of the same vesicle co-labeled with **e** VAMP2-pHmScarlet (magenta) and VAMP2-SEP (green), **f** VAMP2-pHmScarlet (magenta) and VAMP2-EGFP (green), **g** VAMP2-pHuji (magenta) and VAMP2-SEP (green), or **h** VAMP2-pHmScarlet (magenta) and VAMP2-EGFP (green). Source data is provided as a source data file.

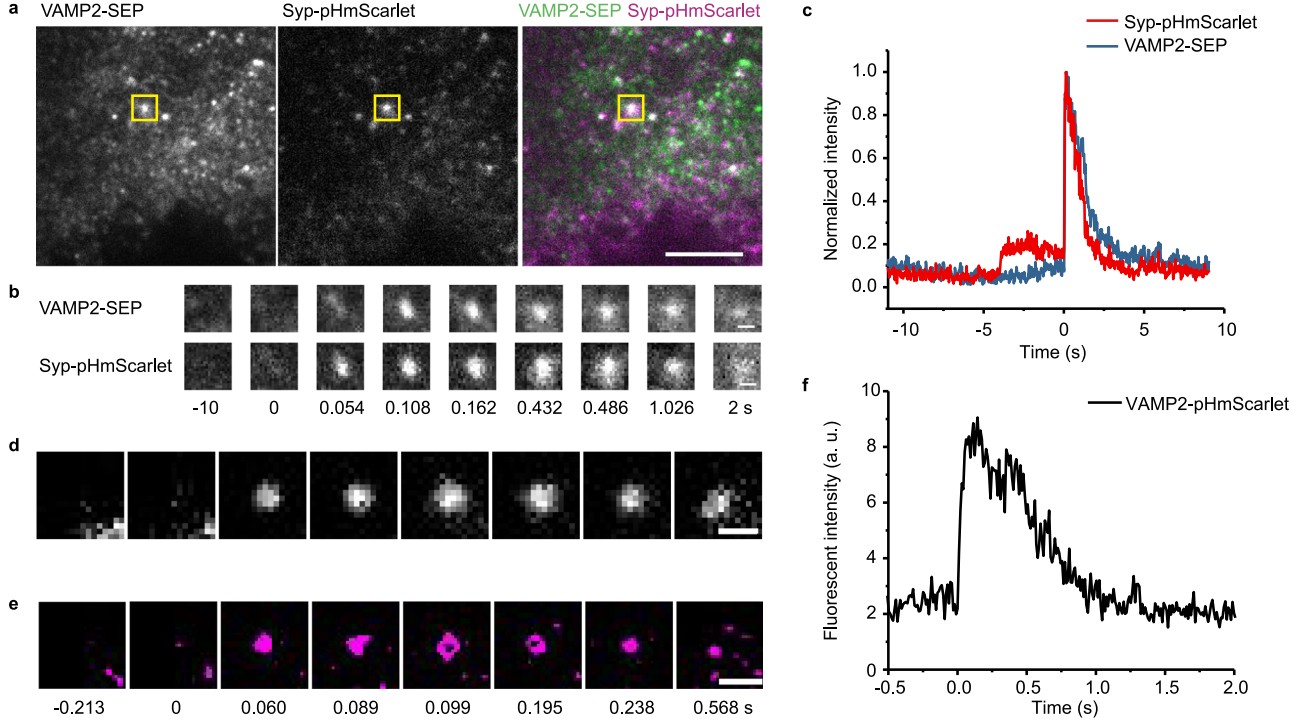

**Fig. 6 Dual-color imaging of two vesicle membrane cargos and super-resolution imaging with pHmScarlet. a** TIRF images of INS-1 cells expressing VAMP2-SEP and Syp-pHmScarlet. The locations of vesicle fusion are indicated by yellow rectangles, scale bar: 5 μm. Experiments were repeated three times. **b** Time-lapses of the marked exocytotic events (bottom, 1.3 × 1.3 μm), scale bars: 0.5 μm. **c** Normalized intensity traces of the marked event in the green and red channels of the same vesicle. **d** Wide-field TIRF image and **e** montages of a representative fusion event with ring structure in a VAPM2-pHmScarlet-labeled vesicle (magenta). Experiments were repeated five times for (**d**) and (**e**). **f** Intensity traces of the marked event shown in (**e**).

docking and fusion steps of single-vesicle exocytosis were analyzed using the time-lapse fluorescence-intensity curves, whereas the vesicle structure was determined at different time points using Hessian-SIM and time alignment. There are 12% of the fusion events (40 of 331) exhibited ring structures. The results illustrated in Fig. 6e demonstrate that pHmScarlet-labeled vesicles have a ring structure similar to that of SEP-labeled vesicles[16]. This structure cannot be observed using wide-field TIRF microscopy (Fig. 6d). The images recorded over time show that the unique ring structures of pHmScarlet-labeled vesicles correspond to opened fusion pores (Fig. 6d–f).

## Discussion

Both pH-sensitive FPs and dyes are important imaging tools to compensate each other for monitoring exocytosis. Previously Martineau et al. developed "semisynthetic" red pH-sensitive protein conjugates (CFI and VO) with organic fluorophores that can be combined with SEP for dual-color imaging[10]. CFI and VO exhibit higher photostability and brightness than pHuji, but additional experimental procedures such as long-term loading (several hours) and washing steps are required using these non-fluorogenic dyes to avoid the potential problem of background signal of unbound dyes. In the current study, we have developed pH-sensitive red FPs pHmScarlet0 and pHmScarlet that have high brightness comparable to VO (Supplementary Table 1). As fusion tags, pHmScarlet0 and pHmScarlet are easy for genetic manipulation and have high labeling specificity and potential application for direct imaging of vesicle secretion in deep tissues. Notably, pHmScarlet enables visualization of both vesicle docking and fusion, and SR imaging of vesicle exocytosis, which have not been observed and reported in Martineau et al.'s paper.

Although the pH sensitivity of exocytosis-tracking FPs is strongly dependent on pKa and $n_H$, brightness and photostability are also important characteristics of high-performance fluorescent proteins. In this study, the brightest available red FP (mScarlet-I) was chosen as a template for the preparation of two red FPs that are highly sensitive to physiological pH, namely pHmScarlet0 and pHmScarlet. pHmScarlet has a nearly optimal pKa (7.4) and a high $n_H$ value (1.1). Its fluorescence intensity increases by ∼26 times when pH is increased from 5.5 to 7.5. Comparatively, pHmScarlet0 exhibits lower pH sensitivity and $n_H$ value; however, it is two times more photostable than pHmScarlet (Table 1), and so, it is suitable for long-time imaging of exocytosis in neuronal and endocrine secretory cells.

Compared to pHuji, a recently developed red pH-sensitive FP, pHmScarlet is almost six times brighter and much more photostable. Moreover, it does not exhibit the photo-switching behavior observed in pHuji. As such, pHmScarlet is more suitable for long term and SR imaging of secretion events, and it has great potential for the development of quantitative exocytosis imaging methods (e.g., in synaptic terminals[20]) based on photobleaching (e.g., FRAP[21]). The FP engineered in this study also enables simultaneous monitoring of the docking and fusion steps by detecting changes in the protein's fluorescence intensity. Considering that pHmScarlet fluoresces at around 589 nm, it can be combined with SEP for dual-color imaging, unlike pHorans which exhibits spectral bleed-through.

High $n_H$ and optimal pKa values are required for the observation of fusion events since they are associated with high burst signals. However, FPs with less than ideal values of $n_H$ and pKa can also be useful in certain applications. For example, although pHmScarlet has a lower $n_H$ than SEP, its nearly ideal pKa renders it effective in the detection of vesicle docking, a process that

cannot be observed using SEP labeling. In general, it is desired that a pH-sensitive FP be useful in monitoring both, the docking and fusion steps of exocytosis. This ability is governed by two main factors: the fluorescence intensity of the FP in the docking stage and its sensitivity to pH changes during fusion pore opening. EGFP and other red pH-sensitive FPs like mOrange[22] (pKa 6.5) retain much of their fluorescence in the intact vesicle, and so, they have great potential for tracking and imaging the docking step prior to fusion. However, these FPs cannot be used ideally to detect fusion due to the fact that they are insufficiently pH sensitive and have a high background signal from the vesicles near the plasma membrane. In contrast, SEP has high pH sensitivity and is nearly non-fluorescent inside intact acidic vesicles. Therefore, it is ideal for the detection of vesicle fusion during exocytosis, but not suitable for monitoring the docking step. As for pHmScarlet, it combines the advantages of EGFP and SEP. With a sufficiently strong signal in the intact vesicle and high pH sensitivity, pHmScarlet is capable of simultaneously detecting the docking and fusion steps. At pH 5.5, the fluorescence intensity of pHmScarlet is high due to its bright nature (almost six times brighter than pHuji) and nearly optimum pKa. Therefore, the docking step can be clearly observed using pHmScarlet. In a previous study, it had been demonstrated that mApple and EGFP are also capable of simultaneously detecting vesicle docking and fusion when combined. However, when these two FPs are used together, the two most commonly used channels are occupied, which impedes the imaging of other biological processes occurring during single-vesicle exocytosis. Besides, mApple has much lower pH sensitivity than pHmScarlet and pHuji.

In summary, the developed red pH-sensitive FP (pHmScarlet) shows high performance in monitoring single-vesicle exocytosis in INS-1 cells, HT-22 hippocampal neuronal cells, and primary hippocampal neuron cells. pHmScarlet is characterized by nearly optimal pKa and relatively high $n_H$ value, as well as good brightness and photostability. Therefore, it has great potential in numerous applications pertaining to neurons and other secretory cells. For example, pHmScarlet can be used to study the dynamics of docking and fusion, as well as the evolution of other regulating proteins at the release site, during exocytosis. Such data is essential for the elucidation of the molecular mechanisms implicated in the exocytosis process. pHmScarlet is also compatible with existing green FP-based functional probes such as calcium indicators, and so, it can be used for dual color functional imaging.

## Methods

**Mutagenesis and screening of red pH-sensitive FPs.** The cDNA of mScarlet-I was optimized, synthesized, and cloned in the pRSETa vector. The plasmid was used as the initial template for the construction of genetic libraries by site-directed saturation mutagenesis. The mutagenesis experiments were performed on the BL21 strain of *Escherichia coli* (AlpaLife, Shenzhen, China), based on the method of polymerase incomplete primer extension (PIPE). The transformed cells were grown overnight and at 37 °C on LB-agar supplemented with 50 μg mL$^{-1}$ ampicillin (Sigma). Subsequently, single colonies were picked and inoculated into 4 mL LB medium containing 50 μg mL$^{-1}$ ampicillin. The mixtures were cultured overnight, then the proteins from a fraction of the cell pellet were extracted using the bacterial protein extraction kit (CoWin Biosciences, Beijing, China), according to the manufacturer's guidelines. The pH sensitivity of the extracted FPs was analyzed using a Varioskan Flash spectral scanning multimode reader (Thermo Scientific). The fluorescence intensities were measured in buffers of pH 5.5 and 7.5.

**Protein expression and purification.** To purify the pH-sensitive FPs, the heat-shock method was used to transform the chemically competent *Escherichia coli* strain BL21 with the plasmid of interest. The bacterial cells were cultured overnight on LB-agar plates containing 50 μg mL$^{-1}$ ampicillin, then single colonies were picked and grown in 4 mL LB at 37 °C. Afterward, an expansion culture was carried out in 200 mL ampicillin-supplemented LB until the optical density reached 0.6. Protein expression was induced by adding 0.8 mM isopropyl β-D-1-thiogalactopyranoside (IPTG) and growing the culture overnight at 16 °C. The bacteria were harvested at 10,000 × $g$ and 4 °C for 10 min, then they were lysed using a cell disruptor. The lysed cells were subsequently centrifuged at 12,000 × $g$

and 4 °C for 15 min, and the proteins were collected from the supernatant using the Ni-NTA His-Bind resin (Qiagen), according to the manufacturer's instructions. The collected proteins were purified by gel filtration using a Superdex 200 Increase 10/300 GL column (GE Healthcare), then they were concentrated and diluted in PBS (Gibco, Carlsbad, CA, USA).

**Characterization of FPs in vitro.** Absorption spectra of the extracted FPs were recorded on an Agilent 8453 UV-visible spectrophotometer using 3.5 mL quartz cuvettes (Starna Cells) with a path length of 1 cm. Meanwhile, an F7000 fluorescence spectrophotometer (Hitachi, Japan) was used to acquire the emission and excitation spectra. In all 630 and 510 nm light were used for excitation and emission spectroscopy measurements, respectively. The fluorescence quantum yields and molar extinction coefficients of the FPs were calculated relative to the reported values of mScarlet (quantum yield: 0.7; molar extinction coefficient at 588 nm: 100 M$^{-1}$ cm$^{-1}$).

The pH sensitivity curves were recorded using a Varioskan Flash spectral scanning multimode reader (Thermo Scientific). For this purpose, the purified FPs were diluted with the desired pH buffer (pH 3 to 11) in 96-well clear-bottomed plates. The pKa (pH at which fluorescence emission reaches 50% of maximum) and apparent $n_H$ values of each FP were determined by fitting the normalized data to the following Eq. (1):

$$F = \frac{1}{1 + 10^{n_H(pKa - pH)}} \qquad (1)$$

The photobleaching rates were determined according to a previously published method[23]. In brief, purified FPs (same concentration) were embedded in polyacrylamide, and their time-lapse fluorescence intensities were measured using an Olympus IX-71 system equipped with a 100× objective (NA = 1.49; adjustable iris 0.5–1.25) oil lens. The fluorescence of different FP samples was recorded for 160 s under identical conditions (excitation wavelength: 561 nm; exposure time: 50 ms; energy density of laser: 31.032 W cm$^{-2}$), and the data were fit to a double exponential function using the Origin software.

**Photochromic behavior measurement.** U-2 OS cells were transfected with H2B-FPs. After 36 h, the samples were fixed and the photochromic behavior was analyzed using widefield microscopy. Briefly, at zero time point, the cells were illuminated with a 561 nm laser. The excitation light was manually alternated between 561 nm at 77.8 W cm$^{-2}$ and 488 nm at 33.3 W cm$^{-2}$ or 405 nm at 5.5 W cm$^{-2}$ every 2 s for multiple illumination cycles. The mean intensity was determined per cell and the background was subtracted. The mean intensity immediately after ($I_2$) and before ($I_1$) 488 or 405 nm induced switching were used to calculate the photochromic behavior. The photochromic amplitude (Ph chr) was calculated using equation (2): Ph chr = $\frac{I_2 - I_1}{I_2}$ × 100%. Ph chr: photochromic amplitude, $I_1$: mean intensity immediately before 488 or 405 nm light, $I_2$: mean intensity immediately after 488 or 405 nm light.

**Plasmid constructs.** VAMP2-SEP and Syp-SEP were gifted by Dr. Liangyi Chen of Peking University (Institute of Molecular Medicine, China). VAMP2-pHmScarlet and VAMP2-pHuji were created by replacing the cDNA of SEP in the plasmid VAMP2-SEP with that of pHmScarlet or pHuji. To construct Syp-pHmScarlet, SEP in the Syp-SEP plasmid was replaced with pHmScarlet. The H2B cDNA was reverse transcribed from the mRNA of HEK293T cells and cloned in the NheI/XhoI sites of pEGFP-N1 (Addgene) in order to produce the H2B-EGFP vector. The SNAP-tag cDNA was amplified by PCR and fused with the C terminal of H2B-EGFP to produce the H2B-EGFP-SNAP-tag vector. H2B-pHmScarlet-SNAP-tag and H2B-pHuji-SNAP-tag were created by replacing the cDNA of SEP with either pHmScarlet or pHuji. To construct the plasmid for the OSER assay, the cDNAs of pHmScarlet0 and pHmScarlet were amplified with a 5′ primer including an Age I site and a 3′ primer containing a Not I site. The PCR products were gel purified, digested, and ligated to Age I and Not I-treated pCytERM-EGFP cloning vector. To prepare FP-α-tubulin plasmids, the cDNA of α-tubulin was amplified from cDNA library and inserted into the vector of pEGFP-C1 digested by Sal I and BamH I, and then pEGFP-α-tubulin was digested with Nhe I and Bgl II to replace EGFP with mScarlet, pHmScarlet0, or pHmScarlet. Primers used for plasmid construction were listed in Supplementary Table 2. All plasmids were sequenced (RuiBiotech, Beijing, China) before further analysis. The restriction enzymes were purchased from Thermo Fisher Scientific.

**Cell culture and transfection.** INS-1 cells were cultured in RPMI 1640 Medium (GIBCO, 11835-030) supplemented with 10% FBS, 1 mM sodium pyruvate solution, 50 μM β Mercaptoethanol (GIBCO, 21985023), and 0.1% Mycoplasma prevention reagent (Transgen) at 37 °C and 5% CO$_2$. The HT-22 hippocampal neuronal cells were maintained in DMEM (Invitrogen) containing 10% FBS in an incubator at 37 °C and 5% CO$_2$. U-2 OS cells were culture in McCoy's 5 A Medium Modified (MCMM) with 10% FBS and cultured in a humidified incubator kept at 37 °C and 5% CO$_2$. As for the primary hippocampal neurons, rat hippocampi were dissected from P0 Sprague-Dawley and were dissociated for 15 min with 0.25% trypsin in Hank's balanced salt solution (37 °C), without Ca$^{2+}$ and Mg$^{2+}$. Then, the hippocampi were triturated in DMEM supplemented with

10% F12 and 10% FBS (Gibco). The hippocampal neurons were plated on 25 mm poly-D-lysine-coated coverslips and placed in 35 mm dishes ($1.0–1.2 \times 10^5$ cells per dish). After 4 h of plating, the medium was replaced with serum-free Neurobasal A (NB-A) solution containing 2% B27 supplement and GlutaMAX (Gibco, Carlsbad, CA, USA). Half the medium volume was replaced every 3 days until use.

Cell line and primary neuronal transfections were carried out using Lipofectamine 2000 and Lipofectamine LTX (Invitrogen, Carlsbad, CA, USA), respectively, as per the manufacturers' instructions. The neuronal transfections were performed on 10–12 days in vitro (DIV) after plating. Briefly, the DNA (2.0 μg well$^{-1}$) was mixed with 2.0 μL PLUS reagent in 200 μL Neurobasal A medium, then 4.0 μL Lipofectamine LTX in 200 μL NB-A medium were added. The mixtures were incubated for 20 min before being added to the neurons in NB-A at 37 °C and 5% $CO_2$. After 1 h, the neurons were rinsed with NB-A and incubated in the original medium at 37 °C and 5% $CO_2$ for 4–5 days.

**Characterization of the monomeric properties of FPs**. For the OSER assay[18,19], FPs were fused to the cytoplasmically oriented ER Membrane (CytERM) and expressed in U-2 OS cells for 36 h, and then images were taken using the fast super resolution confocal laser microscope LSM980 (Zeiss). The percentage of cells with a normal reticular-shaped ER and without incorrect localization, was counted. The ratio of the averaged intensity of the whorls structure (OSER) over the mean intensity of the nuclear envelope (NE) was measured in cells expressing different CytERM-FP constructs. For visualization of microtubules, U-2 OS cells were transfected with 500 ng plasmid. After 36 h transfection, cells were imaged using the same microscopy.

**Live cell imaging**. The INS-1 and HT-22 cell lines were maintained in a buffer solution containing 136 mM NaCl, 4.2 mM KCl, 1.2 mM $MgSO_4$, 2.4 mM $CaCl_2$, 4 mM D-glucose, 10 mM HEPES, and 1 mM L-glutamine (pH 7.4 with NaOH). To induce vesicle exocytosis, the cells were transferred to a stimulation buffer containing 70 mM NaCl, 70 mM KCl, 2.4 mM $CaCl_2$, 1.2 mM $KH_2PO_4$, 1.2 mM $MgSO_4$, 15 mM glucose, 10 mM HEPES, and 1 mM L-glutamine (pH 7.4 with KOH). In order to detect the vesicle exocytosis of primary neuronal cells, the neurons were washed with 25 mM HEPES buffer containing 119 mM NaCl, 2.5 mM KCl, 2 mM $CaCl_2$, 2 mM $MgCl_2$, 30 mM glucose, 10 μM 6-cyano-7-nitroquinoxaline-2,3-dione (CNQX), and 50 μM DL-2-amino-5-phosphonovaleric acid (DL-AP5) (pH 7.4). The stimulation buffer had the same composition except that the KCl concentration was elevated to 90 mM and the NaCl concentration adjusted to 31.5 mM to keep ionic strength constant. The temperature was maintained at 37 °C using a stage incubator and an objective warmer (H301-K-FRAME and OBJ-COLLAR-2532, OKOLAB, Italy). Images were acquired using a homemade TIRF microscopy system with an Olympus IX71 body (Apo N 60×/1.49 NA oil objective, Olympus, Japan).

For dual-color imaging, 488 and 561 nm lasers (Sapphire 488LP-200 and Sapphire 561LP-200, Coherent, USA) were jointly used. The lasers were coupled into an optical fiber (QPMJ-3S3S-488-3.5, OZ, Canada), then they were collimated by an objective lens (CFI Plan Apochromat Lambda 2× NA 0.10, Nikon, Japan). An acousto-optic tunable filter (AOTF, AA Opto-Electronic, France) was used to control the two lasers. The issued light was passed through an AC508-300-A lens (Thorlabs, USA) and a multiband dichroic mirror (Di01-R405/488/561/635-25 × 36, Semrock, USA) in order to focus on the back focal plane of the objective lens. As for the emitted light, it passed through a filter cube holding the dichroic mirror and a multiband filter (FF01-446/523/600/677-25, Semrock) before being separated by a dichroic mirror (560 DCXR, Semrock) on the image splitter (Optosplit II, Cairn, UK). The separated light was then projected through filters (525/50 and 607/70, Semrock) onto the chip of an EM-CCD camera (iXon Ultra 897, Andor, UK). The images were acquired sequentially at alternate 488 and 561 nm excitation and 50 or 100 ms exposure (primary neuron cells) using the Andor Solis software. The red and green images were aligned post acquisition using projective image transformation. Before conducting the experiments, 100 nm fluorescent microspheres (blue/green/orange/dark red, T7279, Invitrogen) were imaged in the green and red channels, then they were superimposed by mapping the bead positions.

The procedure of Hessian-SIM SR imaging is detailed in a previous study[16]. The analysis was conducted with the Hessian-SIM software. In brief, the images were acquired by TIRF-SIM based on a commercially available inverted fluorescence microscope (IX83, Olympus) equipped with a TIRF objective (Apo N 100×/1.7 NA oil objective, Olympus) and a 561 nm laser. We applied a frame rate of 97 Hz in live SIM imaging. Using rolling reconstruction, the raw data acquisition rate was increased to 291 Hz (with exposure and transition times of 0.5 and 0.62 ms, respectively).

**Ethics statement**. All animal experiments were approved by and performed in accordance with the guidelines of the Animal Care and Use Committee of Institute of Genetics and Developmental Biology, Chinese Academy of Sciences (Approval code: AP2013003 and AP2015002).

**Reporting summary**. Further information on research design is available in the Nature Research Reporting Summary linked to this article.

## Data availability

Any data that support the findings of this study beyond what is included in the Supplementary Information are available from the corresponding author upon request. Requests for unique biological materials such as plasmids should be directed to the corresponding author. Mammalian expression plasmids of pHmScarlet0 and pHmScarlet are available at Addgene. Source data are provided with this paper.

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

## Acknowledgements

This project was supported by the National Key R&D Program of China (2016YFA0501500 and 2017YFA0505300), the National Natural Science Foundation of China (31421002, 21778069, 31970704, 31901061, 21927813, 31530039, and 91954126), the Strategic Priority Research Program of Chinese Academy of Sciences (XDB37040301), the Project of Chinese Academy of Sciences-Peking University Leading Cooperation Team, the National Laboratory of Biomacromolecules, the Clinical Medicine Plus X—Young Scholars Project, Peking University, the Fundamental Research Funds for the Central Universities, and the High-performance Computing Platform of Peking University. We thank Dr. Lei Wang (Institute of Biophysics, Chinese Academy of Sciences) for the purified Ero1α. We thank LetPub (www.letpub.com) for its linguistic assistance during the preparation of this manuscript.

## Author contributions

P.X. designed research. A.L. performed the screening and evolution of pHmScarlet. A.L., L.Y., and W.H. performed imaging experiments. F.X. participated in data analysis. Y.Y. performed the primary neuronal experiments. W.H. set up the TIRF imaging microscope. X.H. and W.H. set up the SIM imaging microscope. L.C. and J.L. participated in discussing on the project. P.X. and L.Y. analyzed the data and wrote the paper.

## Competing interests

The authors declare no competing interests.
