## [Peer Review File · Nature Communications]

Reviewers' Comments:

Reviewer #1:

Remarks to the Author:

Liu et al. Describe the generation of pHarlet, a pH sensitive red fluorescent protein evolved from mScarlet-I. pHarlet is reported to have a high pK of 7.4, good photostability and no fast photo-switching and it shows a >20 fold contrast in the pH 5.5-7.5 range. The authors demonstrate they can observe both vesicle-docking and -fusion with pHarlet in cultured cells and neurons. Pharlet clearly outperforms existing red pH-probes and is comparable in detecting membrane fusions as the gold standard in the field SEP. Together, it presents two major advancements in the field: firstly, unlike SEP it allows visualization of both vesicle docking and fusion, and secondly it allows dual color imaging of multiple vesicle docking events by using SEP and pHarlet-labeled proteins. I think the paper represents an important advance in the field, yet I have the following concerns of which 1) is really a major issue:

1) The mutations K163L, A165V of Pharlet (introduced in the pHarlet0 or pHarlet-KLV variant, see line 114) are both facing to the exterior of the beta-barrel. These two amino acids form the center of the so-called AC-interaction interface of tetrameric RFPs such as DsRed and eqFP611. While the AB-interface of tetrameric RFPs is easily broken with only one or two mutations, extensive mutagenesis of RFPs was necessary to prevent dimerization along the AC-interface. For instance, in mRFP1, the first monomeric RFP evolved from DsRed, two mutations H163K and A165R were introduced (together with many other changes) to break the AC dimerization interface. For mScarlet-I K163 was maintained. Also the hydrophobic T148C mutation is situated very nearby 163L, 165V, together making a hydrophobic patch, likely re-establishing the AC-interaction dimer interface. Note that in the AC-dimer, the other monomer is sandwiched exactly around these same 3 amino acids, so these 3 amino acids reinforce themselves in the other monomer of this dimerization interface. Moreover, since T148 of adjacent monomers in the AC interface are directly facing each other at a distance of less than 5 Angstrom, I even do not exclude the formation of a disulfide-bridge in the oxidizing milieu of the luminal side of the endomembrane compartments (such as the ER) between two pHarlet beta-barrels at C148. Especially this T148C mutation should have been discarded in the further evolution of the cysteine-free pHarlet0 variant. So I strongly suspect that pHarlet(0) is dimeric (perhaps even covalent in the lumen of endomembrane compartments), and that K163L, A165V not only realign K164 (as suggested in line 112), but that they establish a dimer interface, which contributes to the higher pK and the higher photostability. If pHarlet is indeed dimeric, this has implications for using it as fusion tag, especially for membrane proteins for which it may cause non-physiological dimerization. In order to rule out dimerization, the authors should perform an OSER test (see Costantini et al, Traffic 2012; 13: 643-649) and make a straight fusion of pHarlet to alpha-tubulin to demonstrate microtubule structures. Disulfide-bridge formation should be checked by running a gel of purified pHarlet under oxidative and reducing environment (e.g. with and without beta-mercaptoethanol or DTT).

2) In the spectroscopic characterization of pHarlet, Figure 1a) pH-dependent excitation spectra are shown. Also, pH-dependent absorbance spectra should be shown. In addition to the decrease of the absorbance at 570 nm, it is likely that a clear increase can be seen in the blue absorption around 450 nm, representing the protonated chromophore. Using these spectra, the in vitro pK of chromophore protonation can be determined.

3) In figure 1e it is shown that no fast off-switching occurs for pHarlet (in contrast to pHuji). In line 135-136 it is described that no photoswitching occurs. Many red fluorescent proteins show reversible photochromicity. For instance mApple (also harboring the crucial pK changing M164K mutation) can be reversibly switched on and off for >50% by alternating blue and yellow excitation. This process involves protonation and deprotonation coupled to cis-trans isomerization of the chromophore. In this paper, the authors claim there is no photoswitching observed for pHarlet, but in my opinion they only demonstrate an apparent lack of fast off-switching. To demonstrate that pHarlet is non-photochromic, they should test the effect on fluorescence intensity of alternating blue (say 450-480 nm) excitation and regular yellow/green excitation (540-560 nm) and the lack of fast photochromicity. This issue of photochromicity does not appear to be interfering with single-color imaging using yellow light excitation, but, depending on the results of

blue/yellow light induced photoswitching, it can seriously interfere with the green/red two channel imaging of SEP and pHarlet. So to aid future biologists who would use pHarlet in dual color experiments, a note on how to circumvent photochromicity-related artefacts should be included, if indeed blue/yellow photochromicity turns out to be substantial for pHarlet under alternating blue/yellow illumination conditions.

Reviewer #2:

Remarks to the Author:

In this manuscript, authors report a new FP, pHarlet, to study exocytic vesicle fusion and docking. Although, the protein can be of interest for some researchers, in my opinion the results shown here do not warrant publication in Nature Communication. There are two main reasons:

1- They generated pHarlet, which is not as sensitive to pH as SEP. Therefore, authors use this "shortcoming" and claim this could be a good FP for studying docking. They present it as a strategy to study docking but eventually it is an FP that is not as good as SEP for studying exocytosis.

2- Even though I can accept that this FP can be used to study docking, to be publishable in Nat Comm, I expect to see something solved with this probe that we did not know before. Authors rarely show any biological data, everything they show is just proof-of-principle.

Therefore, I think this paper is more suitable for journals like Communication Biology.

My detailed comments are below..

Authors do not mention important work that actually addressed the lack of red fluorescent tags for exocytosis. For instance, in Martineau et al, they showed pH-sensitive protein conjugates as a red alternative of SEP that can be combined with SEP, enabling dual colour imaging. Moreover, this technology allows for labelling with fluorophores that are brighter and more photostable, allowing super-resolution imaging. <https://www.nature.com/articles/s41467-017-01752-5>

"This suggests that pHarlet is more sensitive than SEP and pHuji for the detection of single-vesicle exocytosis." How is this possible if SEP is twice more sensitive to pH according to Table 1? How can $\Delta F_{max}/F_0$ be higher for pHarlet?

Also, it is not clear to me how authors calculated $\Delta F_{max}/F_0$. How can this ratio be 0 (which means no contrast in burst?) Does number of labelled protein changes this ratio?

How many experiments does Fig2g represent? Please add information and statistics.

Did authors compare pHarlet with other FPs in neuronal cells?

"It should be noted that only 83.6% of the pHarlet-labeled vesicles were found to be docked at the plasma membrane. The remaining vesicles skip the docking stage and enter the fusion step directly" this might be very misleading. Maybe pHarlet cannot capture the rest of the docking event. Therefore, the data is not strong enough to talk about "docking-free exocytosis".

In the kymographs, I can see gradual increase in fluorescence for pHarlet (docking), but in figure 2, I see a very sudden increase in emission. Don't these contradict?

Authors should refer to each panel of each figure in the text (such as Fig4a,b). Also, it is confusing that sometimes further panels are mentioned before the earlier panels.

This sentence requires reference: "Likewise, pHuji cannot be used to reliably monitor the docking and fusion kinetics of vesicle exocytosis due to its photo-switching behavior"

"As expected, the mScarlet-IKLV mutant (K163L and A165V) termed pHarlet 0 (pH-sensitive mScarlet 0) presents increased pH sensitivity compared to mScarlet-IK. The pKa of pHarlet 0 is 7.2, and it shows a 20-fold change in fluorescence intensity upon increasing the pH from 5.5 to 7.5." Where is this data? Please refer to a figure or a table.

In the abstract, there are many unsubstantiated statements which read a bit strange. I know they explain this throughout the manuscript but still, reader should understand the abstract clearly since most of the readers these days only read the abstract.

"However, the docking step cannot be visualized using this FP." Why not?

"Among the available red pH-sensitive FPs, none is comparable to SEP for practical application" what are the practical reasons?

Reviewer #3:

Remarks to the Author:

In this work, the authors describe a novel pH-sensitive red fluorescent protein (FP), which the name pHarlet, with properties that surpass existing pH-sensitive red FPs in terms of pKa, nh of the fluorescence against pH curve, fluorescence brightness and photostability for the detection of exocytosis events. The characteristics of pHarlet encompass the advantages of both EGFP and superecliptic pHluorin (SEP) where its pKa value enables the detection of vesicle docking, with detectable fluorescence intensity at acidic conditions in intact vesicles, and its nh value provides appropriate sensitivity to pH changes to distinguish fusion events. They further demonstrated the utility of pHarlet by performing dual-color imaging of two vesicle membrane cargos utilizing pHarlet and SEP-labeled vesicle cargos and super-resolution imaging of Vamp2-pHarlet in INS-1 cells. Overall, the manuscript is well written and the data look solid supporting pHarlet as a potentially beneficial tool for the visualization of both vesicle docking and fusion processes during exocytosis.

There are several issues that could be addressed to strengthen the work. One point that seems trivial, but may be worth thorough consideration is the name. While it makes some sense as the protein was derived from ScarletFP, the resonance of pHarlet is rather unfortunate in the English language. This point by no means affects scientific judgement of this work, but terminology can play a role in its adoption by the community.

The other major issue is that some bits of basic information on pHarlet were not provided. While the characteristics of pHarlet are shown in different situations, the manuscript must include the excitation and emission spectra plots of pHarlet 0 along with the complete DNA sequences of pHarlet 0 and pHarlet. In addition, a schematic of the mutation sites in mScarlet for the generation of both pHarlet 0 and pHarlet should be provided at least in the supplement.

A number of details could be included or clarified:

- In Figure 1, Table 1 and the Methods section, the photobleaching rate, $t_{1/2}$, was determined by fitting the fluorescence decay curve with a double exponential function. Which of the decay constants was reported as $t_{1/2}$ and why? Also, in Table 1, it would be helpful to have the extinction coefficient and quantum yield for SEP as well, either reporting what is available in the literature or by direct measurement as they have the necessary equipment.
- For the comparison of the number of fusion events among pHarlet, SEP and pHuji in Fig. 2g, it would be more accurate to compare the data by normalizing the number of fusion events with respect to cell area and time.
- The authors performed dual-color imaging to monitor the exocytosis of two distinct vesicle membrane cargos, synaptophysin-pHarlet and VAMP2-pHarlet, in INS-1 cells. However, the derived conclusion of the similar diffusive behavior of the two labeled cargos based on the similar decay kinetics of pHarlet and SEP seemed rather lacking as the authors concluded their findings solely on a single exocytosis event. Moreover, the capability of pHarlet to track docking events was not highlighted in the result.
- Despite the various examples that illustrate the biological application of pHarlet in Figure 4, quantitative analyses of the vesicle docking/fusion events and super-resolved structures are missing.
- Line 105: It is stated, 'In a previous study, it had been shown that residue 163 in pHuji...', no citation was provided.
- Line 161: The authors state that a wider distribution of burst suggests higher sensitivity for detection of single fusion events. This seems inaccurate. They are referring to fig 2B, which shows possibly a bimodal distribution. One can speculate on the causes of that, but it is not obvious how that means a higher sensitivity to single events. The authors should clarify this point and explain it better in the text.
- Line 224: It is stated that difficulty in attributing the cause of a first peak in the pHuji trace

suggests pHuji cannot be used reliably. However, this seems to be related to that specific image rather than a feature of pHuji, unless they are saying that that is always the case when colabeling vesicles with pHuji.

- Lines 268 and 270, references to the figure appear to be mislabeled, and not correspond to Figure 4.

POINT-BY-POINT RESPONSE TO REVIEWER'S COMMENTS

Reviewer #1:

Liu et al. Describe the generation of pHarlet, a pH sensitive red fluorescent protein evolved from mScarlet-I. pHarlet is reported to have a high pK of 7.4, good photostability and no fast photo-switching and it shows a >20 fold contrast in the pH 5.5-7.5 range. The authors demonstrate they can observe both vesicle-docking and -fusion with pHarlet in cultured cells and neurons. pHarlet clearly outperforms existing red pH-probes and is comparable in detecting membrane fusions as the gold standard in the field SEP. Together, it presents two major advancements in the field: firstly, unlike SEP it allows visualization of both vesicle docking and fusion, and secondly it allows dual color imaging of multiple vesicle docking events by using SEP and pHarlet-labeled proteins. I think the paper represents an important advance in the field, yet I have the following concerns of which 1) is really a major issue:

- 1) the mutations K163L, A165V of pHarlet (introduced in the pHarlet0 or pHarlet-KLV variant, see line 114) are both facing to the exterior of the beta-barrel. These two amino acids form the center of the so-called AC-interaction interface of tetrameric RFPs such as DsRed and eqFP611. While the AB-interface of tetrameric RFPs is easily broken with only one or two mutations, extensive mutagenesis of RFPs was necessary to prevent dimerization along the AC-interface. For instance, in mRFP1, the first monomeric RFP evolved from DsRed, two mutations H163K and A165R were introduced (together with many other changes) to break the AC dimerization interface. For mScarlet-I K163 was maintained. Also the hydrophobic T148C mutation is situated very nearby 163L, 165V, together making a hydrophobic patch, likely re-establishing the AC-interaction dimer interface. Note that in the AC-dimer, the other monomer is sandwiched exactly around these same 3 amino acids, so these 3 amino acids reinforce themselves in the other monomer of this dimerization interface. Moreover, since T148 of adjacent monomers in the AC interface are directly facing each other at a distance of less than 5 Angstrom, I even do not exclude the formation of a disulfide-bridge in the oxidizing milieu of the luminal side of the endomembrane compartments (such as the ER) between two pHarlet beta-barrels at C148. Especially this T148C mutation should have been discarded in the further evolution of the cysteine-free pHarlet0 variant. So I strongly suspect that pHarlet(0) is dimeric (perhaps even covalent in the lumen of endomembrane compartments), and that K163L, A165V not only realign K164 (as suggested in line 112), but that they establish a dimer interface, which contributes to the higher pK and the higher photostability. If pHarlet is indeed dimeric, this has implications for using it as fusion tag, especially for membrane proteins for

which it may cause non-physiological dimerization. In order to rule out dimerization, the authors should perform an OSER test (see Costantini et al, Traffic 2012; 13: 643–649) and make a straight fusion of pHarlet to alfa-tubulin to demonstrate microtubule structures. Disulfide-bridge formation should be checked by running a gel of purified pHarlet under oxidative and reducing environment (e.g. with and without beta-mercaptoethanol or DTT).

Response:

Thank you so much for your comments and suggestions. As suggested, the organized smooth endoplasmic reticulum (OSER) approach was performed. For better determination of OSER structures, EGFP was included as a positive control for dimerization, and mScarlet was included as a negative control as it was stated to be a true monomer. pHarlet0 and pHarlet showed the same performance as monomeric mScarlet in U-2 OS cells, with over 89% of cells showing normal phenotypes (Supplementary Figure 5). Moreover, α -tubulin was labeled with EGFP, mScarlet, pHarlet0 or pHarlet. The expressions of mScarlet, pHarlet0, and pHarlet-tagged α -tubulin resulted in bright microtubules with minor background fluorescence, supporting the results that like mScarlet, both pHarlet0 and pHarlet are monomeric in living cells (Supplementary Figure 6). We also checked the disulfide-bridge formation by running a gel of purified pHarlet 0 and pHarlet under oxidative and reducing environment, respectively. Ero1a was included as a positive control and mScarlet was included as a negative control. Under oxidative environment, Ero1a migrated faster than that under the reducing treatment, which is consistent with previously results (J. Biol. Chem. 2014, 289:31188-31199), whereas, no significant difference was observed in the oxidative and reducing group for mScarlet, pHarlet0 and pHarlet, respectively (Supplementary Figure 7).

Supplementary Figure 5: Assessment of oligomeric state of FPs in living cells by OSER assay. U-2 OS cells were transfected with plasmids encoding fusions of the cytoplasmically oriented ER Membrane (CytERM) to different FPs: (a) mScarlet, (b) pHmScarlet0, (c) pHmScarlet, and (d) EGFP. Scale bar is 10 μm . (e) Ratio of the averaged intensity of the whorls structure (OSER) over the mean intensity of the nuclear envelope (NE) was measured in cells expressing different CytERM-FP constructs. Each dot represents an OSER structure. The center values and the error bars in the graph represent mean value and error bars (SD). (f) Table displaying the results of the OSER assay. ‘Normal looking cells’ are cells without OSER structures and without incorrect localization.

Supplementary Figure 6: Microtubule structure in living cells using Airyscan SR microscopy. U-2 OS cells were transfected with plasmids encoding fusion constructs with FP-7aa- α -tubulin: (a) mScarlet- α -tubulin, (b) pHmScarlet0- α -tubulin, (c) pHmScarlet- α -tubulin, and (d) EGFP- α -tubulin. Scale bar is 10 μm .

Supplementary Figure 7: Identification of the oligomeric states of pHmScarlet0 and pHmScarlet under oxidative and reducing environment. (a) purified mScarlet, pHmScarlet0, and pHmScarlet were analyzed by nonreducing SDS-10% PAGE under oxidative (without DTT/ β -ME, line1, mScarlet; line2, pHmScarlet0 and line3, pHmScarlet) and reducing environment (with DTT/ β -ME, line 4, mScarlet; line5, pHmScarlet0 and line 6, pHmScarlet). No significant difference was observed in the oxidative and reducing group for mScarlet, pHmScarlet0 and pHmScarlet, respectively. (b) Ero1 α was included as a positive control. Purified Ero1 α was loaded using buffer with or without DTT/ β -ME and analyzed by nonreducing SDS-8% PAGE. Under oxidative environment, Ero1 α migrated faster (lane 7) than that of under the reducing treatment (line 8). M: marker, NR: nonreducing, R: reducing.

We think there may be some explanations for the monomeric property of pHarlet. We totally agree that H162 and A164 of DsRed are important to form the AC-interaction interface. But as you said, “extensive mutagenesis of RFPs was necessary to prevent dimerization along the AC-interface”, besides H162 and A164, other 7 amino acid are involved in the evolution of mRFP0.1 from DsRed (PNAS, 2002, 99, 12:7877-7882), which indicated that these residues together with H162 and A164 to determine the AC-interaction interface. Furthermore, the

templet of pHarlet is mScarlet-I, which sequence is very different from that of mRFP, especially those amino acids facing to the exterior of the beta-barrel (D161, R167, R173, D177, K183). Thus, we think K163L and A165V are not enough to reform AC-interaction interface back as did in DsRed, and the four amino acids (D161, R167, R173, D177, highlighted in blue in the lower panel of Response Figure 1) may play a key role in breaking the hydrophobic patch formation in DsRed.

	10	20	30	40	50							
mScarlet-I	MVSKGEAVI	KEFMRFKV	HMEGSMNG	HEFEIEEGE	GEGRPYEGT	QTAKLKVT : 50						
pHmScarlet	MVSKGEAVI	KEFMRFKV	HMEGSMNG	HEFEIEEGE	GEGRPYEGT	QTAKLKVT : 50						
DsRed	-MRSSKNV	IKVKEFMR	FKVRMEG	TVNGHEFEI	EEGEGEGR	PYEGHNTV : 49						
mRFP1	-MASSE	DAVIKEF	MRFKV	RMESV	NGHEFEIE	EEGEGEGR : 49						
	60	70	80	90	100							
mScarlet-I	KGGPLPFS	WDILSPQ	FMYGSR	AFI KHPAD	IPDYYKQ	SFPEGFKW : 100						
pHmScarlet	KGGPLPFS	WDILSPQ	FMYGSR	AFI KHPAD	IPDYYKQ	SFPEGFKW : 100						
DsRed	KGGPLPFA	WDILSPQ	FQYGS	KVYV	KHPADIP	DYK KLSF : 99						
mRFP1	KGGPLPFA	WDILSPQ	FQYGS	KAYV	KHPADIP	DY LKLSF : 99						
	110	120	130	140	150							
mScarlet-I	EDGGAVT	VTQDTS	LEDGTL	IYKVKL	LRGTN	FPPDGPV : 150						
pHmScarlet	EDGGAVT	VTQDTS	LEDGTL	IYKVKL	LRGTN	FPPDGPV : 150						
DsRed	EDGGVVT	VTQDSS	LQDGG	CFIYK	VKFI	GVNFP : 149						
mRFP1	EDGGVVT	VTQDSS	LQDGG	EFIYK	VKLR	GTNFP : 149						
	160	170	180	190	200							
mScarlet-I	L YPEDG	V LKGD	I KMA	LRLK	DGGRY	L ADFK	TTYK	A KKP	VQMP	GAYN	VDR	KL : 200
pHmScarlet	L YPEDG	V LKGD	I LKVL	RLRK	DGGRY	L ADFK	TTYK	A KKP	VQMP	GAYN	VDR	QL : 200
DsRed	L YPRD	G V LK	G E I H	K A L	K L K	D G G	H Y L	V E F	K S I	Y M	A K K	P V Q L : 199
mRFP1	M Y P E	D G A	L K G	E I K	M R L	K L K	D G G	H Y D	A E V	K T T	Y M	A K K : 199
	210	220										
mScarlet-I	DITSHNE	DYTVV	EQYER	SEGRH	STGG	MDELYK- : 232						
pHmScarlet	TITSHNE	DYTVV	EQYER	SEGRH	STGG	MDELYK- : 232						
DsRed	DITSHNE	DYTI	VEQYER	TEGRH	HLFL	----- : 225						
mRFP1	DITSHNE	DYTI	VEQYER	AEGRH	STGA	----- : 225						

Response Figure 1: Upper: sequence alignment of the amino acids of mScarlet-I, pHmScarlet (the name “pHarlet” in the first version of manuscript), DsRed and mRFP1. The evolution of DsRed to mRFP1 are highlighted in dark blue boxes. Light blue boxes indicated the amino acid mutations from mScarlet-I to pHmScarlet. Different amino acids between DsRed and pHmScarlet are highlighted in Red boxes. Lower: simulated crystal structure of pHmScharlet referring to that of mScharlet (PDB No. 5LK4). Magenta indicates residues 163L and 165V. Blue indicates the different amino acids facing to the exterior of the beta-barrel and near 163L and 165V between DsRed and pHmScarlet. We think the four amino acids (D161, R167, R173, D177) may play a key role in breaking the hydrophobic patch formation in DsRed.

- 2) In the spectroscopic characterization of pHarlet, Figure 1a) pH-dependent excitation spectra are shown. Also, pH-dependent absorbance spectra should be shown. In addition to the decrease of the absorbance at 570 nm, it is likely that a clear increase can be seen in the blue absorption around 450 nm, representing the protonated chromophore. Using these spectra, the in vitro pK of chromophore protonation can be determined.

Response:

Absorbance spectra of pHarlet were measured from pH 5 to pH 10 (Supplementary Figure 3).

You are right and as expected, in addition to the decrease of the absorbance around 570 nm, a clear increase can be detected in the blue absorption around 450 nm.

Supplementary Figure 3: Absorbance spectra of pHmScarlet at different pH values.

- 3) In figure 1e it is shown that no fast off-switching occurs for pHarlet (in contrast to pHuji). In line 135-136 it is described that no photoswitching occurs. Many red fluorescent proteins show reversible photochromicity. For instance mApple (also harboring the crucial pK changing M164K mutation) can be reversibly switched on and off for >50% by alternating blue and yellow excitation. This process involves protonation and deprotonation coupled to cis-trans isomerization of the chromophore. In this paper, the authors claim there is no photoswitching observed for pHarlet, but in my opinion they only demonstrate an apparent lack of fast off-switching. To demonstrate that pHarlet is non-photochromic, they should test the effect on

fluorescence intensity of alternating blue (say 450-480 nm) excitation and regular yellow/green excitation (540-560 nm) and the lack of fast photochromicity. This issue of photochromicity does not appear to be interfering with single-color imaging using yellow light excitation, but, depending on the results of blue/yellow light induced photoswitching, it can seriously interfere with the green/red two channel imaging of SEP and pHarlet. So to aid future biologists who would use pHarlet in dual color experiments, a note on how to circumvent photochromicity-related artefacts should be included, if indeed blue/yellow photochromicity turns out to be substantial for pHarlet under alternating blue/yellow illumination conditions.

Response:

Thank you so much for your kind suggestion. This point is very important for dual color imaging of pH sensitive RFPs with SEP. We agree with you that the description of “lack of fast off-switching” is more precise and we changed the description in the revised manuscript. Also, as suggested we checked the photochromic behaviors of pHuji, pHarlet and mScarlet. Besides 561-nm and 488-nm lights were used for alternately exciting the FPs as suggested, we also tested 561/405-nm lights. Negligible photochromic behavior was recorded for pHarlet and mScarlet, while pHuji showed severe photochromic behavior of 36% at 405 nm and 34.4% at 488 nm (Supplementary Figure 4). We added these data in the revision on Line 139-142.

Supplementary Figure 4: Photochromic behavior of pHmScarlet, mScarlet and pHuji measured in mammalian cells. U-2 OS cells were transfected with H2B-5aa-FP. The cells were widefield illuminated with alternating light of 561 nm and 488 nm (a-c) or 405 nm (e-g) light for multiple illumination cycles. According to the equation: $\text{Ph chr} = \frac{I_2 - I_1}{I_2} \times 100\%$, the photochromic amplitudes represented by the blue arrows were calculated. Quantification of photochromic amplitude of RFPs are presented in (d) and (h). (a) and (e): pHmScarlet; (b) and (f): mScarlet; (c) and (g): pHuji. Ph chr: photochromic amplitude, I_1 : mean intensity immediately before 488/405 nm light, I_2 : mean intensity immediately after 488/405 nm light. The center values and the error bars in the graph represent mean value and error bars (SD).

Reviewer #2:

- 1) In this manuscript, authors report a new FP, pHarlet, to study exocytic vesicle fusion and docking. Although, the protein can be of interest for some researchers, in my opinion the results shown here do not warrant publication in Nature Communication. There are two main reasons: They generated pHarlet, which is not as sensitive to pH as SEP. Therefore, authors use this “shortcoming” and claim this could be a good FP for studying docking. They present it as a strategy to study docking but eventually it is an FP that is not as good as SEP for studying exocytosis.

Response:

We agree with you that so far with near-ideal pKa and high n_H , SEP exhibits ideal properties for detecting vesicle exocytosis. However, SEP alone cannot be used for dual color imaging and also cannot visualizing both vesicle docking and fusion events. Furthermore, among the available pH-sensitive red FPs, none is comparable to SEP for practical applications due to unoptimized pH-sensitivity and fluorescence brightness or severe photochromic behavior. The purpose of our study is to develop a pH sensitive red FP. Our experimental results showed that besides the ability of monitoring docking step, pHarlet can also be used for detecting vesicle fusion step as well as SEP: 1) VAMP2-pHarlet and VAMP2-SEP present similar profiles of averaged exocytotic kinetics (Suppl. Fig. 8); 2) Number of exocytosis events detected using pHarlet is comparable to that of SEP (Fig. 2g). Thus, just as Reviewer 1 mentioned, “pHarlet clearly outperforms existing red pH-probes and is comparable in detecting membrane fusions as the gold standard in the field SEP.” Reviewer 3 also think “the characteristics of pHarlet encompass the advantages of both EGFP and SEP where its pKa value enables the detection of vesicle docking, with detectable fluorescence intensity at acidic conditions in intact vesicles, and its n_H value provides appropriate sensitivity to pH changes to distinguish fusion events.”

- 2) Even though I can accept that this FP can be used to study docking, to be publishable in Nat Comm, I expect to see something solved with this probe that we did not know before. Authors rarely show any biological data, everything they show is just proof-of-principle. Therefore, I think this paper is more suitable for journals like Communication Biology.

Response:

Besides developing an urgently needed pH-sensitive red FP in the exocytosis field for dual-color imaging, we also provided many biological data more than just proof-of-principle in our

manuscript. For example: 1) the docking step prior to the sudden burst signal of the fusion step in endocrine cells and neuronal cells has not been reported using SEP and the characteristics and distribution of docking time are not reported; 2) dual color imaging of two different membrane proteins, synaptophysin and VAMP2, in a single vesicle have not been reported. We recorded the secretion of these two membrane proteins in a single vesicle and described their secretion profiles in the manuscript; 3) for the first time, we detected the docking and fusion steps by SR imaging with a pH sensitive red FP.

On one side, we showed strong evidence that pHarlet clearly outperforms existing red pH-probes and is comparable in detecting fusion events as SEP. On the other side, we provided several biological insights using pHarlet. Thus, as Reviewer 1 commented, our manuscript “presents two major advancements in the field: firstly, unlike SEP it allows visualization of both vesicle docking and fusion, and secondly it allows dual color imaging of multiple vesicle docking events by using SEP and pHarlet-labeled proteins.” Our paper “represents an important advance in the field.”

- 3) My detailed comments are below. Authors do not mention important work that actually addressed the lack of red fluorescent tags for exocytosis. For instance, in Martineau et al, they showed pH-sensitive protein conjugates as a red alternative of SEP that can be combined with SEP, enabling dual colour imaging. Moreover, this technology allows for labelling with fluorophores that are brighter and more photostable, allowing super-resolution imaging. <https://www.nature.com/articles/s41467-017-01752-5> “This suggests that pHarlet is more sensitive than SEP and pHuji for the detection of single-vesicle exocytosis.” How is this possible if SEP is twice more sensitive to pH according to Table 1? How can $\Delta F_{max}/F_0$ be higher for pHarlet?

Response:

Actually, in the first version of manuscript, we cited Martineau et al.'s work (Line 66), without comparison to our novel developed pH sensitive probes, considering that the probe we developed belongs to the FP family, not dyes like Martineau. et al. developed. Both FPs and dyes are two types of probes widely used in biological study. FPs and dyes are totally different, but each of them has advantages and limitations. Compared with FPs, dye probes exhibit higher photo-stability and brightness, but produces the background of unbound dyes. More experimental procedures such as loading and washing steps are required using dye probes. Moreover, dye probes are not easy to penetrate deep tissue for in vivo imaging. On the other hand, FPs have zero background of unbound FPs due to the specific genetic labeling, which is

especially suitable for direct imaging of deep tissues. Both FPs and dyes are important imaging tools that compensate for each other.

Now we have added a comparison of the two types of probes in the discussion section of the revised manuscript and cited Martineau et al.'s work again. "Both pH sensitive FPs and dyes are important imaging tools to compensate each other for monitoring exocytosis. Previously Martineau et al developed "semisynthetic" red pH-sensitive protein conjugates (CFI and VO) with organic fluorophores that can be combined with SEP for dual color imaging¹⁰. CFI and VO exhibit higher photo-stability and brightness than pHuji, but additional experimental procedures such as long-term loading (several hours) and washing steps are required using these non-fluorogenic dyes to avoid the potential problem of background signal of unbound dyes. In the current study, we have developed pH sensitive red FPs pHmScarlet0 and pHmScarlet that have high brightness comparable to VO (Supplementary Table 1). As fusion tags, pHmScarlet0 and pHmScarlet are easy for genetic manipulation and have high labeling specificity and potential application for direct imaging of vesicle secretion in deep tissues. Notably, pHmScarlet enables visualization of both vesicle docking and fusion, and SR imaging of vesicle exocytosis, which have not been observed and reported in Martineau et al's paper."

Fluorophore	Emission peak (nm)	Extinction coefficient ($10^3 \text{ M}^{-1} \text{ cm}^{-1}$)	QY (-)	Brightness ($10^3 \text{ M}^{-1} \text{ cm}^{-1}$)	Source
VO	581	90.9	0.40	36.36	Grimm et al.2016
pHmScarlet0	590	86	0.48	41.53	
pHmScarlet	589	85	0.47	39.73	

Supplementary Table 1. In vitro brightness of red pH-sensitive fluorophores

Besides the two key characteristics of pK_a and n_H that determine the sensitivity of pH probes, in the practical imaging in cells, several other factors such as fluorescence intensity and photo-stability of FPs, as well as the background fluorescence produced by the secreted vesicle fusing to the plasma membrane will greatly affect signal-to-noise ratio of recorded fluorescence. Although the n_H value of SEP is higher than pHarlet, the background signal of SEP on the cell surface is much higher than that of pHarlet (the averaged background fluorescence intensity of SEP and pHarlet are 104.53 and 65.20, respectively). Results of Fig.2b, 2d and 2f showed that pHarlet exhibits a wider distribution of fluorescence bursts ($\Delta F_{max}/F_0$, $\Delta F_{max} = F_{max} - F_0$) and a higher averaged $\Delta F_{max}/F_0$ value (Supplementary Figure 8, please also refer to Reviewer 3, Point 8), which indicated that pHarlet has higher signal-to-noise ratio compared to SEP for the detection of single-vesicle exocytosis in cells. In order to describe the results more accurately,

we changed the description to “As pHmScarlet has lower background signal on cell surface than SEP, and is much brighter than pHuji, it exhibits a wider distribution of fluorescence bursts ($\Delta F_{max}/F_0$, $\Delta F_{max} = F_{max} - F_0$) (Fig.2b, 2d and 2f) and a higher averaged $\Delta F_{max}/F_0$ value (Supplementary Figure 8), which indicated that pHarlet has higher S/N (signal-to-noise ratio) compared to SEP and pHuji for the detection of single-vesicle exocytosis in cells.”.

- 4) Also, it is not clear to me how authors calculated $\Delta F_{max}/F_0$. How can this ratio be 0 (which means no contrast in burst?) Does number of labelled protein changes this ratio? How many experiments does Fig2g represent? Please add information and statistics. Did authors compare pHarlet with other FPs in neuronal cells?

Response:

We are sorry for the unclear description of how to calculate $\Delta F_{max}/F_0$. For calculation, the equation: $\Delta F_{max} = F_{max} - F_0$ was used, in which F_{max} is the maximum fluorescence intensity of the peak of exocytosis, F_0 is the local background fluorescence intensity of the vesicle immediately before secretion. We added the description in the revised manuscript (Line 180-182).

Figure 2b, 2d and 2f are the distributions of different $\Delta F_{max}/F_0$ ranges in cells labeled with different pH sensitive FPs. Each column represents a range of $\Delta F_{max}/F_0$. The range of the first column is from 0 to 0.5 and the minimum values of $\Delta F_{max}/F_0$ are 0.29, 0.32, 0.29 for pHarlet, SEP and pHuji, respectively.

According to the equation: $\Delta F_{max}/F_0 = (F_{max} - F_0)/F_0 = F_{max}/F_0 - 1$, the effect of the number of labeled proteins on F_{max} is much higher than that on F_0 , thus we think the number of labelled proteins may change the ratio of $\Delta F_{max}/F_0$. However, the statistic distribution of $\Delta F_{max}/F_0$ will not be affected by the number of labelled proteins, because for different cells, the probabilities of incorporating expressed proteins into different vesicles are unchanged. Furthermore, for fair comparison, we used the same amount of plasmid DNA of pHarlet, SEP and pHuji for transfection.

Three independent experiments were performed for Figure 2g. We added the information and statistics in Figure 2 legend in the revised manuscript (Line 165-166).

We compared pHarlet with SEP and pHuji in neuronal cells, which were shown in the revised manuscript. Similar as the results performed in INS-1 cells, more exocytotic vesicles can be detected using pHarlet and SEP than using pHuji in neuronal cells (Supplementary Figure 9).

Supplementary Figure 9: Detection efficiency of vesicles using SEP, pHmScarlet and pHuji. Equal amount of plasmid DNA of SEP and pHmScarlet or SEP and pHuji were co-transfected into HT22 cells. The detection efficiency for different FPs is calculated by the number of fusion events detected by SEP, pHmScarlet or pHuji in one imaging channel over the total secretion events of two channels. Independent experiments were performed three times, * $p < 0.05$. The center values and the error bars in the graph represent mean value and error bars (SD).

- 5) “It should be noted that only 83.6% of the pHarlet-labeled vesicles were found to be docked at the plasma membrane. The remaining vesicles skip the docking stage and enter the fusion step directly” this might be very misleading. Maybe pHarlet cannot capture the rest of the docking event. Therefore, the data is not strong enough to talk about “docking-free exocytosis”.

Response:

Based on the results showed in Figure 3h, we agree with you that the phenomena of vesicles skip the docking stage and enter the fusion step directly may due to that the docking step cannot be captured using pHarlet, but the other explanation cannot be entirely dismissed, which have already been explained and described in the manuscript (Line258-265): “There exist two possible explanations for such behavior: either the docking process of these vesicles is too fast to be captured, or their intra-environment is too acidic (more so than the vesicles showing docking steps), and thus, the fluorescence of VAMP2-pHmScarlet cannot be monitored. Considering that the docking-free secretion events recorded using VAMP2-pHmScarlet can also be observed in the green channel of the VAMP2-EGFP-labeled vesicles (the pKa of EGFP is 6.0, Fig. 3h), and that the distribution of the docking dwell times is exponential (Fig. 3d), it is believed that the docking step in docking-free secretions is too fast to be captured. However, the other explanation cannot be entirely dismissed.” The description of “docking-free exocytosis” does not mean without docking step exactly, but means the docking step cannot be detected, as we already explained it in detail in the manuscript.

- 6) In the kymographs, I can see gradual increase in fluorescence for pHarlet (docking), but in

figure 2, I see a very sudden increase in emission. Don't these contradict?

Response:

Secreted vesicles could be categorized into two distinct patterns as: "docking-free exocytosis" and "exocytosis with docking". Considering that the existing pH-sensitive probes are used to detect only fusion events, we first chose the vesicles with "docking-free exocytosis" to prove that our probe can detect vesicle secretion as other FPs did, and surpass the existing pH-sensitive red FPs (Figure 2, sudden increase). Then we further highlighted the novel characteristic of pHarlet for detecting docking step during exocytosis in Figure 3 by choosing those exocytotic vesicles with docking steps. There is no contradiction between Figure 2 and Figure 3; while the description of different vesicles without/with docking steps in order in different figures is to increase the logic of the article.

- 7) Authors should refer to each panel of each figure in the text (such as Fig4a,b). Also, it is confusing that sometimes further panels are mentioned before the earlier panels.

Response:

Thank you so much for your comment. We made correction in the revision (Line 285, Line 303 and Line 305).

- 8) This sentence requires reference: "Likewise, pHuji cannot be used to reliably monitor the docking and fusion kinetics of vesicle exocytosis due to its photo-switching behavior"

Response:

Thank you for your suggestion. We added the reference in the revised manuscript (Line 79-80).

- 9) "As expected, the mScarlet-IKLV mutant (K163L and A165V) termed pHarlet 0 (pH-sensitive mScarlet 0) presents increased pH sensitivity compared to mScarlet-IK. The pKa of pHarlet 0 is 7.2, and it shows a 20-fold change in fluorescence intensity upon increasing the pH from 5.5 to 7.5." Where is this data? Please refer to a figure or a table.

Response:

Thank you for your suggestion. The data were shown in Table 1 and we referred them to Table 1 in the revised manuscript (Line 116-117).

10) In the abstract, there are many unsubstantiated statements which read a bit strange. I know they explain this throughout the manuscript but still, reader should understand the abstract clearly since most of the readers these days only read the abstract. “However, the docking step cannot be visualized using this FP.” Why not? “Among the available red pH-sensitive FPs, none is comparable to SEP for practical application” what are the practical reasons?

Response:

Thanks for your suggestion. To make it clear, we modified the abstract as suggested. “Superecliptic pHluorin (SEP), a green pH-sensitive FP, has been widely used for imaging single-vesicle exocytosis. However, the docking step cannot be visualized using this FP, since the fluorescence signal inside vesicles is too low to be observed during docking process.” (Line 27)

“Among the available red pH-sensitive FPs, none is comparable to SEP for practical imaging in living cells due to unoptimized pH-sensitivity and fluorescence brightness or severe photochromic behavior.” (Line 28-29).

Reviewer #3

In this work, the authors describe a novel pH-sensitive red fluorescent protein (FP), which the name pHarlet, with properties that surpass existing pH-sensitive red FPs in terms of pKa, nh of the fluorescence against pH curve, fluorescence brightness and photostability for the detection of exocytosis events. The characteristics of pHarlet encompass the advantages of both EGFP and superecliptic pHluorin (SEP) where its pKa value enables the detection of vesicle docking, with detectable fluorescence intensity at acidic conditions in intact vesicles, and its nh value provides appropriate sensitivity to pH changes to distinguish fusion events. They further demonstrated the utility of pHarlet by performing dual-color imaging of two vesicle membrane cargos utilizing pHarlet and SEP-labeled vesicle cargos and super-resolution imaging of Vamp2-pHarlet in INS-1 cells. Overall, the manuscript is well written and the data look solid supporting pHarlet as a potentially beneficial tool for the visualization of both vesicle docking and fusion processes during exocytosis. There are several issues that could be addressed to strengthen the work.

- 1) One point that seems trivial, but may be worth thorough consideration is the name. While it makes some sense as the protein was derived from ScarletFP, the resonance of pHarlet is rather unfortunate in the English language. This point by no means affects scientific judgement of this work, but terminology can play a role in its adoption by the community.

Response:

As a non-native English speaker, I really appreciate the comment and useful insight. And as suggested we changed pHalet to pHmScarlet in the revision by referring to pHTomato.

- 2) The other major issue is that some bits of basic information on pHarlet were not provided. While the characteristics of pHarlet are shown in different situations, the manuscript must include the excitation and emission spectra plots of pHarlet 0 along with the complete DNA sequences of pHarlet 0 and pHarlet. In addition, a schematic of the mutation sites in mScarlet for the generation of both pHarlet 0 and pHarlet should be provided at least in the supplement.

Response:

Thank you for your useful suggestions. Now we have added the relevant information in the Supplementary data, including the excitation and emission spectra of pHmSarlet0 (Supplementary Figure 2), and a schematic of the mutation sites in mScarlet-I for the generation of pHmSarlet0 and pHmSarlet (Supplementary Figure 1). The complete DNA sequences of

pHmScarlet0 and *pHmScarlet* have been deposited and available on Addgene.

	10	20	30	40
mScarlet-I	M V S K G E A V I K E F M R F K V H M E G S M N G H E F E I E G E G E G R P Y E G T Q T A K L K V T			
mScarlet-IK	M V S K G E A V I K E F M R F K V H M E G S M N G H E F E I E G E G E G R P Y E G T Q T A K L K V T			
pHmScarlet0	M V S K G E A V I K E F M R F K V H M E G S M N G H E F E I E G E G E G R P Y E G T Q T A K L K V T			
pHmScarlet	M V S K G E A V I K E F M R F K V H M E G S M N G H E F E I E G E G E G R P Y E G T Q T A K L K V T			
	60	70	80	90
mScarlet-I	K G G P L P F S W D I L S P Q F M Y G S R A F I K H P A D I P D Y Y K Q S F P E G F K W E R V M N F			
mScarlet-IK	K G G P L P F S W D I L S P Q F M Y G S R A F I K H P A D I P D Y Y K Q S F P E G F K W E R V M N F			
pHmScarlet0	K G G P L P F S W D I L S P Q F M Y G S R A F I K H P A D I P D Y Y K Q S F P E G F K W E R V M N F			
pHmScarlet	K G G P L P F S W D I L S P Q F M Y G S R A F I K H P A D I P D Y Y K Q S F P E G F K W E R V M N F			
	110	120	130	140
mScarlet-I	E D G G A V T V T Q D T S L E D G T L I Y K V K L R G T N F P P D G P V M Q K K T M G W E A S T E F			
mScarlet-IK	E D G G A V T V T Q D T S L E D G T L I Y K V K L R G T N F P P D G P V M Q K K T M G W E A S T E F			
pHmScarlet0	E D G G A V T V T Q D T S L E D G T L I Y K V K L R G T N F P P D G P V M Q K K T M G W E A S T E F			
pHmScarlet	E D G G A V T V T Q D T S L E D G T L I Y K V K L R G T N F P P D G P V M Q K K T M G W E A S T E F			
	160	170	180	190
mScarlet-I	L Y P E D G V L K G D I K M A L R L K D G G R Y L A D F K T T Y K A K K P V Q M P G A Y N V D R K L			
mScarlet-IK	L Y P E D G V L K G D I K K A L R L K D G G R Y L A D F K T T Y K A K K P V Q M P G A Y N V D R K L			
pHmScarlet0	L Y P E D G V L K G D I L K V L R L K D G G R Y L A D F K T T Y K A K K P V Q M P G A Y N V D R K L			
pHmScarlet	L Y P E D G V L K G D I L K V L R L K D G G R Y L A D F K T T Y K A K K P V Q M P G A Y N V D R Q L			
	210	220	230	
mScarlet-I	D I T S H N E D Y T V V E Q Y E R S E G R H S T G G M D E L Y K			
mScarlet-IK	D I T S H N E D Y T V V E Q Y E R S E G R H S T G G M D E L Y K			
pHmScarlet0	D I T S H N E D Y T V V E Q Y E R S E G R H S T G G M D E L Y K			
pHmScarlet	T I T S H N E D Y T V V E Q Y E R S E G R H S T G G M D E L Y K			

Supplementary Figure 1. Amino acid sequence alignment of mScarlet-I, mScarlet-IK, pHmScarlet0 and pHmScarlet. Red box indicated the mutation site from mScarlet-I to mScarlet-IK. Blue boxes represented the mutation residues from mScarlet-I to pHmScarlet0, and pink boxes showed the mutations from pHmScarlet0 to pHmScarlet.

Supplementary Figure 2: Excitation and emission spectra of pHmScarlet0 at pH 7.4.

3) A number of details could be included or clarified:

-In Figure 1, Table 1 and the Methods section, the photobleaching rate, $t_{1/2}$, was determined by fitting the fluorescence decay curve with a double exponential function. Which of the decay constants was reported as $t_{1/2}$ and why? Also, in Table 1, it would be helpful to have the

extinction coefficient and quantum yield for SEP as well, either reporting what is available in the literature or by direct measurement as they have the necessary equipment.

Response:

We apologized for the unclear description. We determined $t_{1/2}$ as the timepoint where the fluorescence intensity was decreased to 50% of the initial fluorescence intensity, referring to the previous paper (Nature Methods, 14, 53–56, 2017). We added the description and citation in the revised manuscript (Line 137-138).

The extinction coefficient and quantum yield of SEP are not available in any literature. So we purified SEP and measured the extinction coefficient and quantum yield as suggested. All the data have been added in Table 1 in the revised manuscript.

- 4) -For the comparison of the number of fusion events among pHarlet, SEP and pHuji in Fig. 2g, it would be more accurate to compare the data by normalizing the number of fusion events with respect to cell area and time.

Response:

As suggested, we normalized the number of fusion events with respect to cell area and time among pHmSarlet, SEP and pHuji, and modified Figure 2g in the revised manuscript.

- 5) -The authors performed dual-color imaging to monitor the exocytosis of two distinct vesicle membrane cargos, synaptophysin-pHarlet and VAMP2-pHarlet, in INS-1 cells. However, the derived conclusion of the similar diffusive behavior of the two labeled cargos based on the similar decay kinetics of pHarlet and SEP seemed rather lacking as the authors concluded their findings solely on a single exocytosis event. Moreover, the capability of pHarlet to track docking events was not highlighted in the result.

Response:

We compared the averaged decay kinetics of pHmSarlet and SEP of exocytotic vesicles co-labeled with synaptophysin-pHmSarlet and VAMP2-SEP and found that the two labeled membrane proteins showed the similar diffusive behavior (Supplementary Figure 11).

Supplementary Figure 11. Averaged normalized intensity traces of syp-pHmScarlet (red) and VAMP2-SEP (blue) vesicle exocytosis in response to high glucose and high $[K^+]$ stimulation in INS-1 cells.

Thanks for the suggestion to highlight the capability of pHarlet to track docking events. Now we replaced Figure 4a, b, and c with another exocytosis event that the docking step can be observed with pHmScarlet.

Figure 4: Dual-color imaging of two vesicle membrane cargos and super-resolution imaging with pHmScarlet. (a) TIRF images of INS-1 cells expressing VAMP2-SEP and Syp-pHmScarlet. The locations of vesicle fusion are indicated by yellow rectangles, scale bar: 2 μm . (b) Time-lapses of the marked exocytotic events (bottom, $1.3 \times 1.3 \mu\text{m}$), scale bars: 0.5 μm . (c) Normalized intensity traces of the marked event in the green and red channels of the same vesicle.

- 6) -Despite the various examples that illustrate the biological application of pHarlet in Figure 4, quantitative analyses of the vesicle docking/fusion events and super-resolved structures are missing.

Response:

We added the quantitative analyses of vesicle docking/fusion events for dual-color imaging and quantified the percentage of super-resolved structures in the revised manuscript, as

“Furthermore, with pHmScarlet the docking steps can be observed and there is appropriate 64.87% of secreted vesicles are with the docking steps.” and “Appropriate 12.08% of the fusion events (40 of 331) exhibited ring structures”.

- 7) -Line 105: It is stated, ‘In a previous study, it had been shown that residue 163 in pHuji...’, no citation was provided.

Response:

We added the citation in the revised manuscript.

- 8) -Line 161: The authors state that a wider distribution of burst suggests higher sensitivity for detection of single fusion events. This seems inaccurate. They are referring to fig 2B, which shows possibly a bimodal distribution. One can speculate on the causes of that, but it is not obvious how that means a higher sensitivity to single events. The authors should clarify this point and explain it better in the text.

Response:

We are sorry for the unclear explanation. In practical imaging in living cells, besides the two key characteristics of pK_a and n_H that determine the sensitivity of pH probes, several other factors such as fluorescence intensity and photo-stability of FPs, as well as the background fluorescence produced by the secreted vesicle fusing to the plasma membrane will greatly affect signal-to-noise ratio of recorded fluorescence. Although the n_H value of SEP is higher than pHmSarlet, the background signal of SEP on the cell surface is much higher than that of pHmSarlet (the averaged background fluorescence intensity of SEP and pHmSarlet are 104.53 and 65.20, respectively). Thus, pHmSarlet exhibits a higher signal-to-noise ratio of single vesicle exocytosis and a wider distribution of fluorescence bursts ($\Delta F_{max}/F_0$, $\Delta F_{max} = F_{max} - F_0$) (Fig.2b, 2d and 2f).

We agree with you that the distribution of fluorescence bursts ($\Delta F_{max}/F_0$) could be described by bimodal distribution, which is consistent with a previously report (Nature Biotechnology, 36,451–459, 2018). We measured the mean $\Delta F_{max}/F_0$ value and found that pHmSarlet has higher averaged $\Delta F_{max}/F_0$ values than that of SEP (Supplementary Figure 8). It is true that the description of “higher sensitivity” is a little bit misleading, so we changed it to “higher signal-to-noise ratio”. Accordingly, we changed the description in the revised manuscript to “As pHmScarlet has lower background signal on cell surface than SEP, and is much brighter than

pHuji, it exhibits a wider distribution of fluorescence bursts ($\Delta F_{max}/F_0$, $\Delta F_{max} = F_{max} - F_0$) (Fig. 2b, 2d and 2f) and a higher averaged $\Delta F_{max}/F_0$ value (Supplementary Figure 8), which indicated that pHarlet has higher S/N (signal-to-noise ratio) compared to SEP and pHuji for the detection of single-vesicle exocytosis in cells.”.

- 9) -Line 224: It is stated that difficulty in attributing the cause of a first peak in the pHuji trace suggests pHuji cannot be used reliably. However, this seems to be related to that specific image rather than a feature of pHuji, unless they are saying that that is always the case when colabeling vesicles with pHuji.

Response:

We colabeled vesicles with pHuji and SEP, and quantitatively analyzed the proportion of secreted vesicles that have only one exocytotic event but with a first peak prior to the fusion peak. We found that appropriate 39.29% of the single exocytotic events were with a small peak, suggesting that nearly 40% of the single exocytotic events are hard to be determined whether it corresponds to a docking event or another fusion event at the same loci when pHuji were used for exocytosis detection. Combined with that 83.60% of secreted vesicles are with docking steps when colabeling vesicles with pHmSarlet and SEP, the data also indicates that nearly 44.31% of the docking events are missed and cannot be detected with pHuji. We added the statistical data in the revised manuscript (Line 248-255).

- 10) -Lines 268 and 270, references to the figure appear to be mislabeled, and not correspond to Figure 4.

Response:

We checked the legends (line 268-270 of the first version of manuscript) and corrected.

Reviewers' Comments:

Reviewer #1:

Remarks to the Author:

The manuscript by Liu et al. has been significantly improved after revision.

All my points regarding possible dimerization have been convincingly addressed. It is clear that despite the suspicious hydrophobic mutations facing the exterior of the beta-barrel at the AC dimerization interface, even including a potentially oxidizable cysteine does not change the monomeric behavior of the pHmScarlet probe, also not in oxidative environments. The OSER experiments, alfa-tubulin-labelling and gel experiment significantly strengthen the entire study, since now a convincing and well-founded statement can be given about the usage of pHmScarlet as monomeric, non-aggregating fusion-tag to probe pH at various cellular locations. Also, my concern with regard to photochromicity is very clearly addressed and in fact, that shows a clear additional advantage of pHmScarlet with negligible photochromicity over pHuji which displays major troubles with a photochromicity of 35%.

I only have some editorial comments:

- 1) Many supplemental figures have been added, also with new experimental conditions. However, the experimental procedures were not updated. I think the manuscript should be updated to include the experimental conditions (e.g., filter sets, microscopy conditions, etc) covering all supplemental figures as well. Possibly, the legends of these supplemental can then subsequently be shortened.
- 2) I would integrate supplemental fig 3 with figure 1, and spend some words on the blue absorbing protonated chromophore that is prominent below pH 7.
- 3) I would strongly suggest to including several supplemental figures as regular figures in the main text. The non-photochromic behavior supplemental fig 4 clearly describes a key advantage over pHuji, why should we read about this in the supplemental info? The monomeric behavior characterization supplemental figs 5 and 6 could be combined as a new figure in the main text. It is an important property of the new probe. In my view that would strengthen the presentation of pHmScarlet as a superior red pH FP probe, and it could stimulate its adoption by the Nature communications readership.
- 4) In lines 250, 289 and 302 the word 'appropriate' is misplaced. It could be deleted. The percentages mentioned should be rounded off to integer numbers, since I do not think the digits behind the comma are statistically significant anyway.

Reviewer #2:

Remarks to the Author:

In the revised version, authors improved their manuscript significantly. Also, they addressed most of my concerns. Although I am still not 100% convinced on the wide applicability of this probe, I appreciate the technical advance and hence, I recommend the publications of this manuscript.

Reviewer #3:

Remarks to the Author:

The Authors have addressed the questions given in our previous critique.

POINT-BY-POINT RESPONSE TO REVIEWER'S COMMENTS

Reviewer #1:

The manuscript by Liu et al. has been significantly improved after revision.

All my points regarding possible dimerization have been convincingly addressed. It is clear that despite the suspicious hydrophobic mutations facing the exterior of the beta-barrel at the AC dimerization interface, even including a potentially oxidizable cysteine does not change the monomeric behavior of the pHmScarlet probe, also not in oxidative environments. The OSER experiments, alfa-tubulin-labelling and gel experiment significantly strengthen the entire study, since now a convincing and well-founded statement can be given about the usage of pHmScarlet as monomeric, non-aggregating fusion-tag to probe pH at various cellular locations. Also, my concern with regard to photochromicity is very clearly addressed and in fact, that shows a clear additional advantage of pHmScarlet with negligible photochromicity over pHuji which displays major troubles with a photochromicity of 35%.

I only have some editorial comments:

- 1) Many supplemental figures have been added, also with new experimental conditions. However, the experimental procedures were not updated. I think the manuscript should be updated to include the experimental conditions (e.g., filter sets, microscopy conditions, etc) covering all supplemental figures as well. Possibly, the legends of these supplemental can then subsequently be shortened.

Response:

Thank you for your comments. The related experimental procedures were added in the revised manuscript (Line 326-334, Line 343-350, Line 370-376, and Line 403-407)

- 2) I would integrate supplemental fig 3 with figure 1, and spend some words on the blue absorbing protonated chromophore that is prominent below pH 7.

Response:

Supplementary figure3 was integrated in figure1, and the description about this figure was added in the revised manuscript (Line 106-107).

- 3) I would strongly suggest to including several supplemental figures as regular figures in the main text. The non-photochromic behavior supplemental fig 4 clearly describes a key advantage over pHuji, why should we read about this in the supplemental info? The monomeric behavior characterization supplemental figs 5 and 6 could be combined as a new figure in the main text. It is an important property of the new probe. In my view that would strengthen the presentation of pHmScarlet as a superior red pH FP probe, and it could stimulate its adoption by the Nature communications readership.

Response:

Thank you for your suggestions. Supplementary figure 4 was moved in the main text and now is regular figure 2 in the revised manuscript. Supplementary figure 5 and figure 6 were combined as figure 3 in the revised manuscript.

- 4) In lines 250, 289 and 302 the word 'appropriate' is misplaced. It could be deleted. The percentages mentioned should be rounded off to integer numbers, since I do not think the digits behind the comma are statistically significant anyway.

Response:

The word 'appropriate' has been deleted in the revised manuscript and the percentages mentioned in the manuscript were modified as integer numbers (Line 195, 197, 198, 200, 216 and 227).

Reviewer #2:

In the revised version, authors improved their manuscript significantly. Also, they addressed most of my concerns. Although I am still not 100% convinced on the wide applicability of this probe, I appreciate the technical advance and hence, I recommend the publications of this manuscript.

Response:

Thank you.

Reviewer #3 (Remarks to the Author):

The Authors have addressed the questions given in our previous critique.

Response:

Thank you.